DOI: 10.1038/s41467-017-01512-5　　OPEN

# Structural insights into *Legionella* RidL-Vps29 retromer subunit interaction reveal displacement of the regulator TBC1D5

Kevin Bärlocher[1], Cedric A.J. Hutter[1], A. Leoni Swart[1], Bernhard Steiner[1], Amanda Welin[1], Michael Hohl[1], François Letourneur[2], Markus A. Seeger[1] & Hubert Hilbi [1]

*Legionella pneumophila* can cause Legionnaires' disease and replicates intracellularly in a distinct *Legionella*-containing vacuole (LCV). LCV formation is a complex process that involves a plethora of type IV-secreted effector proteins. The effector RidL binds the Vps29 retromer subunit, blocks retrograde vesicle trafficking, and promotes intracellular bacterial replication. Here, we reveal that the 29-kDa N-terminal domain of RidL (RidL$_{2-281}$) adopts a "foot-like" fold comprising a protruding β-hairpin at its "heel". The deletion of the β-hairpin, the exchange to Glu of Ile$_{170}$ in the β-hairpin, or Leu$_{152}$ in Vps29 abolishes the interaction in eukaryotic cells and in vitro. RidL$_{2-281}$ or RidL displace the Rab7 GTPase-activating protein (GAP) TBC1D5 from the retromer and LCVs, respectively, and TBC1D5 promotes the intracellular growth of *L. pneumophila*. Thus, the hydrophobic β-hairpin of RidL is critical for binding of the *L. pneumophila* effector to the Vps29 retromer subunit and displacement of the regulator TBC1D5.

[1] Institute of Medical Microbiology, University of Zürich, Gloriastrasse 30, 8006 Zürich, Switzerland. [2] UMR5235, DIMNP, CNRS/Université Montpellier, Place Eugène Bataillon, Montpellier, 34095 cedex 5, France. Correspondence and requests for materials should be addressed to M.A.S. (email: m.seeger@imm.uzh.ch) or to H.H. (email: hilbi@imm.uzh.ch)

ntracellular bacterial pathogens subvert host membrane dynamics to create unique replication-permissive compartments[1]. *Legionella pneumophila*, the causative agent of Legionnaires' disease[2,3], is a facultative intracellular respiratory pathogen, which thwarts the bactericidal endocytic pathway and forms a distinct endoplasmic reticulum (ER)-associated compartment, the *Legionella*-containing vacuole (LCV)[4,5]. LCVs are formed in environmental amoebae, as well as in mammalian macrophages by a largely conserved mechanism[6]. Proteomics of purified pathogen vacuoles from *Dictyostelium discoideum* amoebae or macrophages revealed several small GTPases and large GTPases, many of which are involved in LCV formation[7–10]. The complex process of LCV development implicates the endocytic, secretory, as well as the retrograde vesicle-trafficking pathway[11].

The retrograde vesicle-trafficking pathway promotes cargo transport from endosomal compartments (early, late, and recycling endosomes) to the *trans* Golgi network (TGN) and further through the Golgi apparatus to the ER[12–16]. The pivotal components of this pathway are phosphoinositide (PI)-metabolizing enzymes, e.g., the phosphatidylinositol (PtdIns) 3-kinase (PI3K) Vps34, or the PI 5-phosphatase OCRL (oculocerebrorenal syndrome of Lowe), as well as the retromer complex and sorting nexins (SNXs). Depletion or deletion of components of the retrograde pathway promotes the intracellular replication of *L. pneumophila*, and thus, a functional retrograde pathway restricts the pathogen[17,18].

The retromer coat complex (also termed cargo-selective complex, herein referred to as "retromer") is an evolutionary conserved heterotrimer of the subunits vacuolar protein sorting (Vps) 26, Vps29, and Vps35, the latter of which is binding both Vps26 and Vps29[19–21]. The retromer machinery is assembled through the association of Vps26–Vps29–Vps35 with heterodimers of membrane curvature-inducing SNXs, e.g., SNX1/SNX2 together with SNX5/SNX6[22,23]. The recruitment of the retromer to endosomes requires the interaction of Vps35–Vps26 with activated, GTP-bound Rab7A[24–26]. In turn, the Rab7 GTPase-activating protein (GAP) TBC1D5 binds Vps29 and promotes the release of the retromer[25,27]. Rab5, on the other hand, recruits the PI3K Vps34, thereby locally enriching PtdIns(3)P, which binds to the Phox homology (PX) domain of SNXs[14]. The retromer binds the cytoplasmic part of transmembrane cargo receptors (e.g., acidic hydrolase receptors), followed by the interaction with SNXs, and several accessory proteins participate in retromer-mediated sorting, tubule elongation and stabilization, and vesicle fission and transport[13,14,22,28]. Given the importance of retrograde recycling for cellular homeostasis, it is expected that a number of bacterial pathogens subvert this pathway[11].

*L. pneumophila* determines pathogen–host interactions by means of a small signaling molecule[29], and to a large part through a type IV secretion system (T4SS) termed Icm/Dot (intracellular multiplication/defective organelle trafficking)[30]. This T4SS translocates the amazing number of ~ 300 different so-called "effector" proteins into host cells, where they subvert pivotal processes such as signal transduction, protein production and turnover, as well as membrane and cytoskeleton dynamics[31–34]. Some of the Icm/Dot substrates promote intracellular bacterial replication by targeting host factors such as phytate[35], PI lipids[33,36–39], small GTPases[31,32,40], or the retromer complex[18].

The 131-kDa Icm/Dot substrate RidL binds the retromer subunit Vps29 as well as PtdIns(3)P and impairs retrograde trafficking[18]. The effector protein decreases the amount of retrograde cargo receptors CIMPR or sortilin and SNX1/2 on LCVs, abolishes retrograde trafficking of cholera toxin and Shiga toxin, and promotes intracellular growth and competitive fitness of *L. pneumophila*. The mechanism, by which RidL subverts retromer

function and retrograde trafficking is unknown. Here, we show that the 29-kDa N-terminal domain of RidL ($RidL_{2-281}$) adopts a "foot-like" fold comprising a protruding hydrophobic β-hairpin, which determines the interaction with the Vps29 retromer subunit and displaces the regulator TBC1D5 from the retromer.

## Results

**N-terminal domain of RidL adopts a foot-like structural fold.** Structural information is crucial to understand the function and mechanism of the *L. pneumophila* effector RidL. However, attempts to create a homology model of at least parts of this 131-kDa protein were hampered by the complete lack of structural templates required for model building[41]. Therefore, we attempted to crystallize full-length RidL as well as fragments of the effector protein. Using this approach, we obtained well-diffracting crystals of the N-terminal 29 kDa fragment (aa 2–281) belonging to space group I422 with one molecule in the asymmetric unit (Table 1). The lack of structural templates impeded structure determination by molecular replacement. Therefore, phases were experimentally determined using selenomethionine-labeled protein.

The structure of $RidL_{2-281}$ was solved at 1.9-Å resolution. The electron densities allowed for building a model comprising

**Table 1 Data collection and refinement statistics**

| | $RidL_{2-281}$ (PDB: 5OH5) | $RidL_{10-258}$-Δβ-hairpin (PDB: 5OH6) | $RidL_{2-281}$-Se-Met |
|---|---|---|---|
| *Data collection* | | | |
| Wavelength | 1.00000 | 0.99999 | 0.97852 (peak) |
| Space group | I422 (97) | P2$_1$ (4) | I422 (97) |
| Cell dimensions | | | |
| a, b, c (Å) | 110.14 | 40.10 | 111.24 |
| | 110.14 | 58.05 | 111.24 |
| | 98.51 | 94.42 | 98.39 |
| α, β, γ (°) | 90.00 | 90.00 | 90.00 |
| | 90.00 | 98.28 | 90.00 |
| | 90.00 | 90.00 | 90.00 |
| Resolution range (Å) | 100–1.9 (1.95–1.90) | 50–2.05 (2.10–2.05) | 100–2.60 (2.67–2.60) |
| No. of reflections | 312069 | 183673 | 486483 |
| Unique reflections | 24155 | 26742 | 18174 |
| $R_{meas}$ (%)[a] | 5.5 (195.4) | 6.3 (84.8) | 23.3 (262.7) |
| $I/\sigma_I$ | 29.54 (1.48) | 15.68 (2.32) | 11.98 (1.63) |
| $CC_{1/2}$ (%) | 100.0 (65.5) | 99.9 (88.8) | 99.9 (14.1) |
| Completeness (%) | 100.0 (99.9) | 98.5 (98.6) | 100.0 (100.0) |
| Redundancy | 12.9 | 6.9 | 26.8 |
| | | | |
| *Refinement* | | | |
| Resolution (Å) | 50–1.9 | 50–2.05 | |
| No. reflections (work/test) | 24146/1206 | 26681/1331 | |
| $R_{work}/R_{free}$ (%) | 20.4/21.36 | 20.17/24.22 | |
| No. atoms | | | |
| Protein | 2146 | 3696 | |
| Water | 187 | 125 | |
| B-factor (Å$^2$) | | | |
| Total | 52.5 | 59.2 | |
| R.m.s deviations | | | |
| Bond lengths (Å) | 0.006 | 0.003 | |
| Bond angles (°) | 0.84 | 0.57 | |
| Ramachandran | | | |
| favoured (%) | 97 | 96 | |
| Allowed (%) | 3 | 4 | |
| Outliers (%) | 0 | 0 | |

[a]Values in parentheses are for the last resolution shell

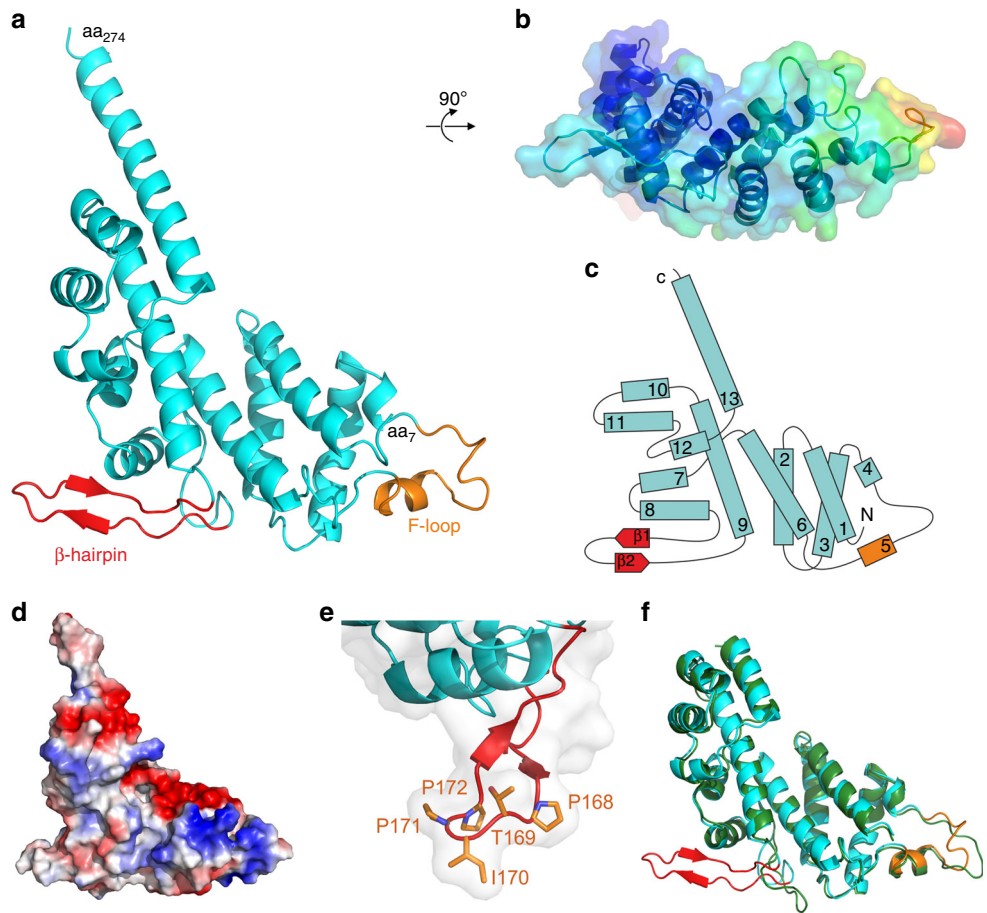

**Fig. 1** The 29 kDa N-terminal domain of RidL adopts a foot-like structural fold. **a** 1.9 Å crystal structure of the 29 kDa N-terminal domain of RidL comprising amino acids 2–281. The β-hairpin (aa 160–180) is shown in red and the F-loop (aa 75–93) in orange. **b** Surface representation of B-factors. Structure from (**a**) depicted in a 90° rotation about a horizontal axis. **c** Schematic representation of the 29 kDa N-terminal domain of RidL, colored as in (**a**). **d** Surface charge distribution (blue for positive, red for negative, white for neutral). The structure is shown in the same orientation as in (**a**). **e** Detail view of the β-hairpin region. $Ile_{170}$ and adjacent residues are shown as sticks. **f** Structural alignment of the 29 kDa N-terminal domain of RidL (colored as in (**a**)) and the corresponding Δβ-hairpin mutant (green) (aa 10–258)

residues 7–274. The structure features a "foot-like" fold, consisting primarily of interacting bundles of α-helices connected by loops of various lengths (Fig. 1a, Supplementary Fig. 1 and Supplementary movie 1). A search for related structures or structural motifs using the Dali server[42] did not reveal any closely related structure deposited in the protein data bank (the nearest structural neighbor with a low Z-score of 4.8 and a high RMSD of 4.4 was PDB entry 1J1J, a human translin protein involved in RNA/DNA binding). The surface representation of B-factors indicates an increasing flexibility from the heel to the toes of the foot (Fig. 1b). At the tertiary structural level, the first eight helices and two β-sheets form the lower part of the foot (Fig. 1c). Helix 9 connects to the foot's ankle, which comprises a perpendicular bundle of three almost parallel helices (10–12). The last helix (13) runs parallel to helix 9, is particularly long, and connects to the remaining, structurally still uncharacterized domains of RidL. Between helices 4 and 6, there is a long loop (aa 75–93) containing three phenylalanines (F-loop) and the short helix 5 (Fig. 1a, c). The F-loop is highly flexible, as judged by the high B-factors of its residues (Fig. 1b).

At the heel of the foot, positioned between helices 8 and 9, there is a surface-exposed hairpin formed by 2 antiparallel β-strands connected by a loop comprising 5 residues (aa 160–180, termed "β-hairpin") (Fig. 1a, c). The β-hairpin is structurally well defined and rigid (Fig. 1b), as well as hydrophobic (based on

surface charge distribution) (Fig. 1d). The β-hairpin and in particular the sequence motif PTIPP at its tip is conserved among *L. pneumophila* RidL homologs. The three prolines in this motif break the antiparallel β-strands and stiffen the tip of the loop. Thereby, the side chain of the conserved $Ile_{170}$ forms an exposed hydrophobic spot (Fig. 1e). Taken together, the 29-kDa N-terminal fragment of RidL features a largely α-helical structural fold comprising a protruding hydrophobic β-hairpin.

**The $RidL_{9-258}$ β-hairpin determines retromer binding in cells.** RidL binds to Vps29, a component of the eukaryotic retromer coat complex (Vps26–Vps29–Vps35)[18]. To determine whether the N-terminal $RidL_{9-258}$ fragment still binds to the retromer, the protein was ectopically produced as a GFP fusion in HeLa cells, and co-localization was assessed by using an antibody recognizing the Vps26 retromer subunit. Fluorescence microscopy showed that $RidL_{9-258}$-GFP co-localized with the retromer in discrete puncta, while the negative control GFP was dispersed throughout the cell (Fig. 2a). The corresponding Pearson's correlation coefficient was approximately 0.7 for $RidL_{9-258}$-GFP and 0.2 for GFP, indicating that co-localization of the former with the retromer was highly significant (Fig. 2b).

A prominent feature of $RidL_{2-281}$ is the exposed hydrophobic β-hairpin comprising amino acid 160–180 (Fig. 1a, c). To exploit

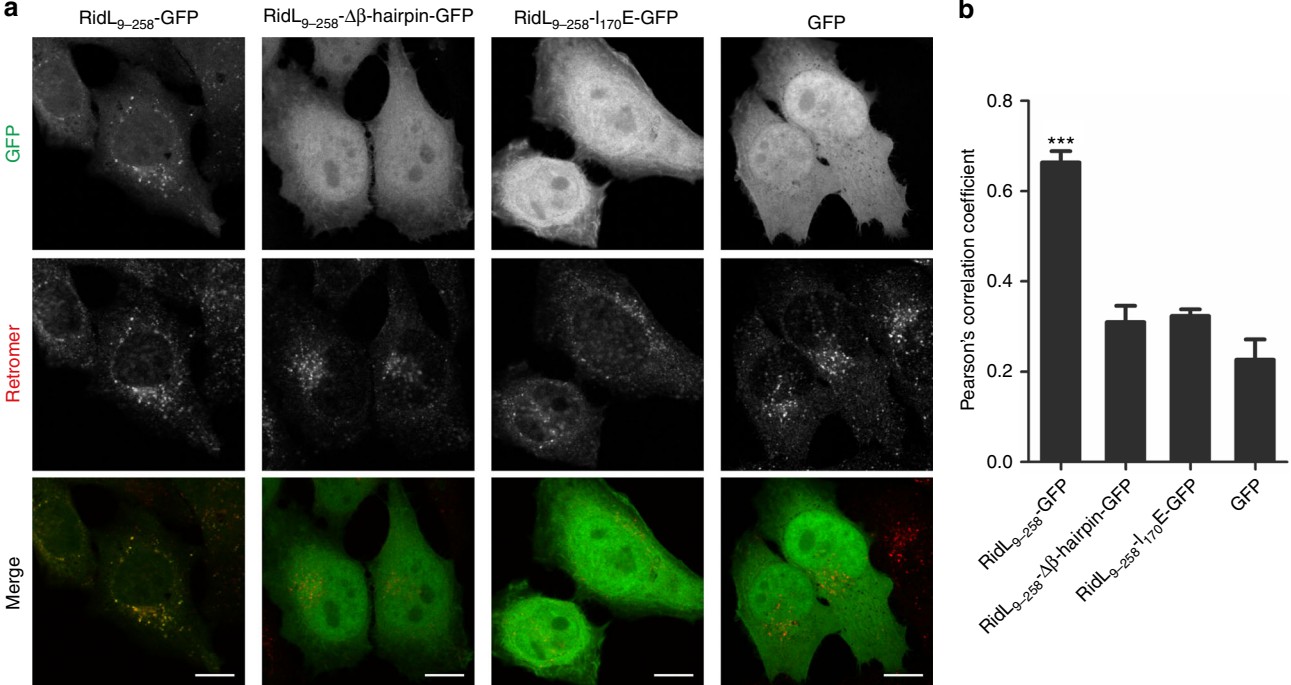

**Fig. 2** The RidL$_{9\text{-}258}$ β-hairpin determines retromer binding of the fragment in cells. **a** HeLa cells were transiently transfected with vectors producing GFP-fusion proteins or GFP as indicated (green) and immuno-stained for the retromer (Vps26; red). Scale bar, 10 μm. **b** Co-localization of RidL variants with retromer (**a**) was scored. The Pearson's correlation coefficient was calculated in three independent experiments with at least 50 cells each (mean and standard deviation, SD, of the mean from each experiment; one way ANOVA; ***$P < 0.001$)

this structural information for functional studies, we tested whether the β-hairpin is involved in retromer binding. To this end, RidL$_{9\text{-}258}$-Δβ-hairpin-GFP was produced in HeLa cells, and co-localization with the retromer was assessed. Interestingly, the fragment lacking the β-hairpin no longer co-localized with the retromer and was found throughout the cell (Fig. 2). This result revealed that the β-hairpin is required for retromer binding and membrane localization of the fragment in general.

To test the possibility that deletion of the β-hairpin leads to global structural changes in RidL, we solved the structure of the N-terminal RidL fragment lacking the β-hairpin at 2.1 Å resolution (Fig. 1f). The N-terminal domain devoid of the β-hairpin crystallized in space group P2$_1$2$_1$2$_1$ with two polypeptide chains (called molecule A and B) in the asymmetric unit (Table 1). These molecules exhibited a root mean square deviation (RMSD) of 0.756 Å over 205 residues among each other, and mainly differ at the F-loop, which was partially undefined due to the high flexibility in molecule A. Superpositions of the Δβ-hairpin mutant with the wild-type RidL domain over the same 205 residues resulted in RMSD of 0.872 Å and 0.962 Å for molecules A and B, respectively (Fig. 1f features the superimposition of molecule B). Thus, the N-terminal RidL fragment lacking the β-hairpin adopts an overall structure that is very similar to the parental fragment (Fig. 1f).

At the very tip of the β-hairpin, the amino acid Ile$_{170}$ is located (Fig. 1e). To test whether this amino acid is involved in retromer binding, a RidL$_{9\text{-}258}$ fragment where Ile$_{170}$ was exchanged to glutamate was produced in HeLa cells, and co-localization with the retromer was assessed. Similar to RidL$_{9\text{-}258}$-Δβ-hairpin-GFP, the fragment RidL$_{9\text{-}258}$-I$_{170}$E-GFP no longer co-localized with the retromer, indicating that Ile$_{170}$ is critical for binding of RidL to the retromer (Fig. 2). The GFP-fusion proteins of RidL$_{9\text{-}258}$, RidL$_{9\text{-}258}$-Δβ-hairpin, and RidL$_{9\text{-}258}$-I$_{170}$E were produced in similar amounts, ruling out protein concentration effects on their cellular localization (Supplementary Figs. 2 and 3). In summary,

the exposed hydrophobic β-hairpin, and in particular the amino acid Ile$_{170}$, determine the binding of RidL$_{9\text{-}258}$ to the eukaryotic retromer.

**The RidL N-terminus binds Vps29 but not Vps29$_{L152E}$ in vitro.** Full-length RidL specifically binds to the retromer subunit Vps29[18], which in turn interacts with the Rab7 GAP TBC1D5 through a hydrophobic pocket including Leu$_{152}$[27]. In initial attempts to study RidL-Vps29 interactions, we used size-exclusion chromatography (SEC). For the SEC experiments, purified RidL (132 kDa), RidL-Δβ-hairpin (130 kDa), or RidL-ΔF-loop (130 kDa) was mixed in a molar ratio of 1:1.5 with Vps29 (21 kDa) and separated with a Superdex 200 column. Comigration of the two proteins was assessed by SDS-PAGE and Coomassie Brilliant Blue staining of fractions eluting from the column (Fig. 3a, Supplementary Fig. 4). This approach confirmed that full-length RidL and RidL-ΔF-loop bind to Vps29, in contrast to the RidL-Δβ-hairpin mutant protein, indicating that the β-hairpin of the N-terminal RidL domain is required for binding to Vps29 in vitro.

Since the mutation Vps29$_{L152E}$ specifically abolishes the interaction with host TBC1D5[43], we further investigated this site for potential binding of RidL. Indeed, while RidL interacted with Vps29, the effector did not bind to Vps29$_{L152E}$ (21 kDa). The SEC analysis of RidL-ΔF-loop or RidL-Δβ-hairpin deletion proteins did not reveal any changes in the elution behavior compared to RidL, indicating that the mutants are correctly folded (Supplementary Fig. 4). In summary, these results confirm that the RidL β-hairpin governs the binding to Vps29, and they reveal that the interaction of Vps29 with RidL is determined by Vps29$_{L152}$, as previously shown for binding of Vps29 to TBC1D5[27].

The interactions of RidL or fragments thereof with Vps29 or Vps29$_{L152E}$ were also analyzed by surface plasmon resonance (SPR). To this end, biotinylated RidL was immobilized on a SPR

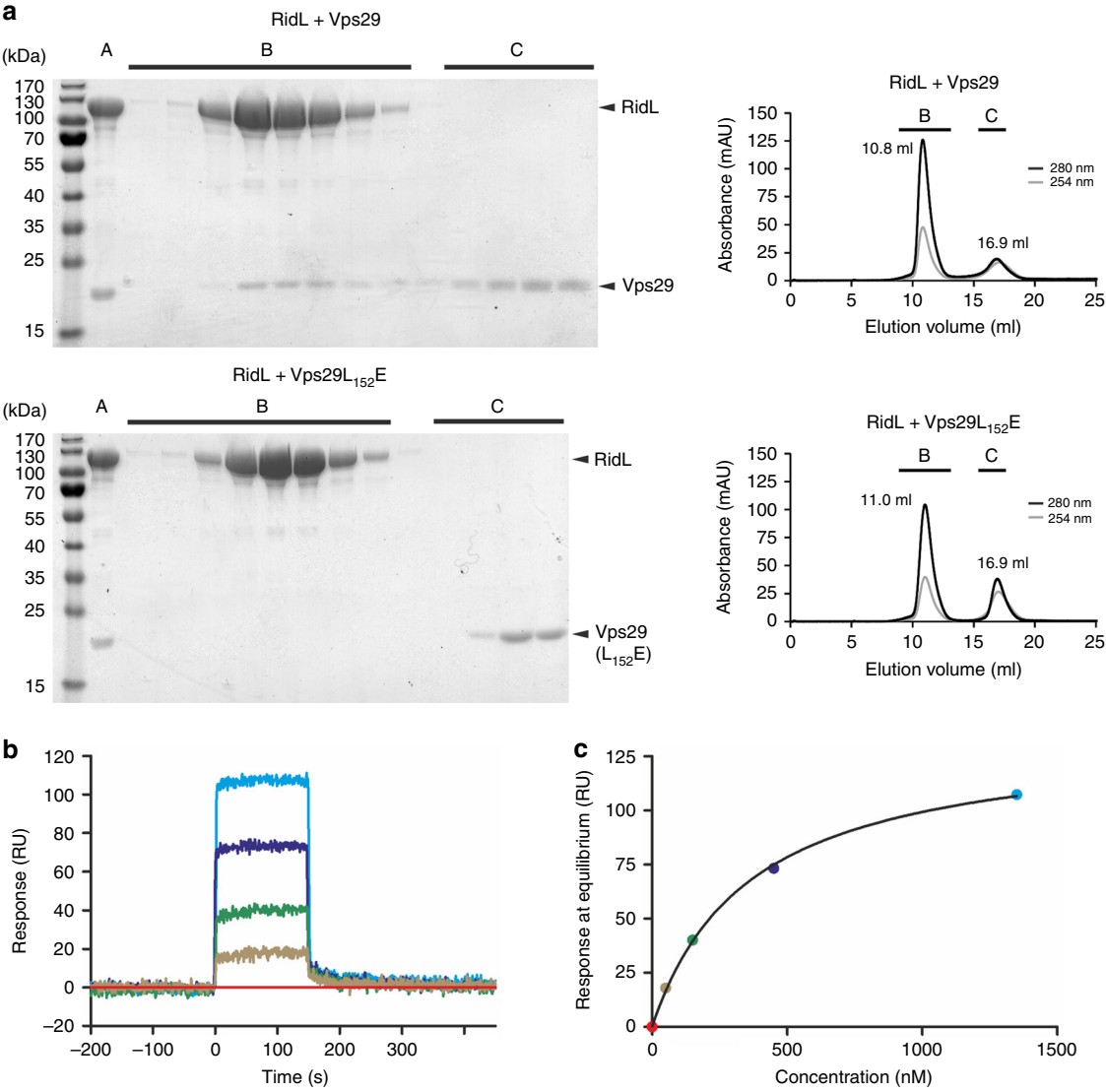

**Fig. 3** RidL binds Vps29 but not Vps29$_{L152E}$ in vitro. **a** For size-exclusion chromatography (SEC), RidL (132 kDa) was mixed in a molar ratio 1:1.5 with Vps29 (21 kDa) or Vps29$_{L152E}$ (21 kDa). SDS-PAGE and Coomassie Brilliant Blue staining of (A) input (ratio 1:1.5), (B) SEC elution fractions 9–13 ml (elution volume RidL or complex), and (C) SEC elution fractions 15.5–17.5 ml (elution volume Vps29 or Vps29$_{L152E}$) were analyzed. SEC elution profiles are shown, analyzed fractions are labeled and elution peak elution volumes are indicated. The peak shift for the RidL–Vps29 complex is 0.2 ml relative to RidL alone. **b** For surface plasmon resonance (SPR) measurements, biotinylated RidL was used as ligand and Vps29 as analyte. Concentration series: red, 0 nM; brown, 50 nM; green, 150 nM; blue, 450 nM; cyan, 1.35 µM. **c** Response units (RU) obtained at binding equilibrium of each Vps29 concentration shown in **b** are plotted against concentration (nM). An equilibrium constant $K_D$ of 400 nM was calculated from the fit as indicated in the Materials and Methods section

chip and purified Vps29 served as the analyte (Fig. 3b). The rates of binding and dissociation were extremely fast, suggesting that the interaction is diffusion-limited. The fast apparent association rate ($> 10^6\,M^{-1}\,s^{-1}$) cannot be accurately determined by SPR due to mass transport limitations, and therefore, the corresponding resonance curves were not fitted to determine on- and off-rates of Vps29 binding. By contrast, binding equilibria levels determined at varying Vps29 concentrations are unaffected by mass transport limitation and permitted to determine a dissociation constant $K_D$ of 400 nM (Fig. 3c).

In the reciprocal SPR approach, biotinylated Vps29 or Vps29$_{L152E}$ were immobilized on the SPR chip, and various RidL variants served as analytes (Supplementary Fig. 4). The resulting SPR curves could only be analyzed qualitatively, because the binding signals did not saturate within the set of analyzed RidL concentrations (Table 2). Binding of full-length RidL$_{2–1167}$ to wild-type Vps29 was confirmed, demonstrating that this

interaction is detectable, regardless of which of the two proteins is immobilized on the SPR chip. The N-terminal 29-kDa RidL$_{10–258}$ fragment binds with a similar affinity to Vps29 as the full-length protein, as judged qualitatively from the SPR sensorgrams (Supplementary Fig. 4). The differences in the absolute response units of the sensorgrams can be fully explained by the different masses of the analytes (RidL: 132 kDa, N-terminal fragment: 29 kDa). The finding that the N-terminal domain is solely responsible for binding to Vps29 was corroborated with the observation that the C-terminal fragment RidL$_{259–1167}$ did not show any detectable binding.

To confirm the potential Vps29 interaction site within the N-terminal domain of full-length RidL, variants lacking either the F-loop (RidL$_{2–1167}$-ΔF-loop) or the β-hairpin (RidL$_{2–1167}$-Δβ-hairpin) were tested. The ΔF-loop mutant protein still exhibited the binding to Vps29 in SPR experiments, similar to what was observed by SEC (Supplementary Fig. 4 and Table 2). However, in

SPR experiments, the maximal response ($R_{max}$) was much decreased, as compared to wild-type RidL, indicating that the F-loop may play some role in the binding to Vps29. As expected, the Δβ-hairpin mutant did not reveal any detectable binding to Vps29 (Supplementary Fig. 4 and Table 2). Neither purified full-length RidL nor any of the fragments exhibited a detectable binding to the Vps29$_{L152E}$ mutant (Supplementary Fig. 4 and Table 2), indicating that RidL binds to the same hydrophobic region of Vps29 as TBC1D5[25,27,43]. Taken together, dependent on the β-hairpin, the RidL retromer-interacting fragment binds to

Vps29 but not to Vps29$_{L152E}$. Therefore, the protruding hydrophobic β-hairpin of RidL and the hydrophobic pocket around Leu$_{152}$ of Vps29 determine the interaction of RidL with the retromer subunit.

**Information-driven docking of RidL and Vps29.** In this study, we show that RidL binds via its hydrophobic β-hairpin into the same hydrophobic pocket of Vps29 as does TBC1D5, since single-point mutations either at the tip of the RidL β-hairpin (I$_{170}$E) or at the center of the hydrophobic cavity of Vps29 (L$_{152}$E) were sufficient to completely abolish the interaction (Figs. 2, 3, and Supplementary Fig. 4). Recently, the crystal structure of Vps29 in complex with a peptide of TBC1D5 was determined[27]. The structure revealed that the peptide binds to a hydrophobic groove at the surface of Vps29 and contains a leucine residue (Leu$_{142}$), whose aliphatic side chain points deeply into a narrow cavity formed by residues Leu$_2$, Leu$_{25}$, Val$_{27}$, Leu$_{152}$, and Tyr$_{165}$ of Vps29.

Based on these findings, we reasoned that Ile$_{170}$ of RidL might bind into the tight cavity of Vps29 in an analogous fashion as Leu$_{142}$ of TBC1D5. Taking this restraint into account, we docked RidL and Vps29 using the webserver HADDOCK2.2. We found a highly scored solution, in which the β-hairpin of RidL interacts with Vps29 over a total surface of 903 Å² with the side chain of Ile$_{170}$ deeply buried in the cavity (Fig. 4a). Importantly, the C$_\alpha$

**Table 2 SPR analysis of RidL–Vps29 interactions**

| Analyte | Mol. Mass (kDa) | Binding to | |
| --- | --- | --- | --- |
| | | Vps29 | Vps29$_{L152E}$ |
| RidL$_{2-1167}$ | 132 | Yes | No |
| RidL$_{10-258}$ | 29 | Yes | No |
| RidL$_{259-1167}$ | 102 | No | No |
| RidL$_{2-1167}$-Δβ-hairpin | 130 | No | No |
| RidL$_{2-1167}$-ΔF-loop | 130 | Yes | No |

For SPR measurements, biotinylated Vps29 or Vps29$_{L152E}$ were immobilized, and RidL variants were used as analytes. Due to limited signal saturation within the measured analyte concentration range, binding was assessed only qualitatively (Supplementary Fig. 4). The RidL β-hairpin and a Vps29 hydrophobic groove determine protein-protein interaction

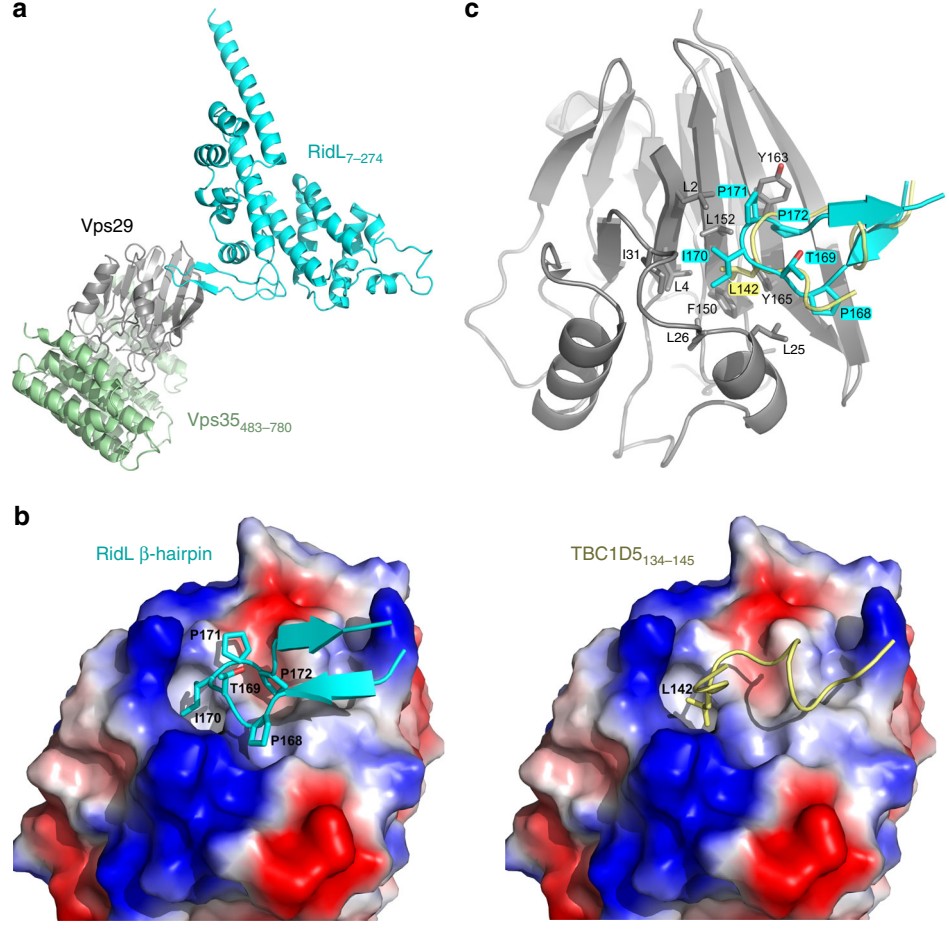

**Fig. 4** Information-driven docking of RidL and Vps29. **a** Docking of RidL (cyan) and Vps29 (grey), in relation to the location of Vps35 (green). **b** The β-hairpin of RidL (cyan) extensively interacts with a hydrophobic groove at the surface of Vps29 (left). The same groove accommodates a peptide of TBC1D5 (right, PDB: 5GTU). The electrostatic potential of the Vps29 surface is shown (red and blue indicate negative and positive charge, respectively). **c** Close-up view of the RidL–Vps29 interaction site. Side chains lining the hydrophobic Vps29 (grey) cavity are highlighted. The β-hairpin of RidL (cyan) is shown as cartoon with sticks. As a reference, the interacting residues of TBC1D5 (yellow) are shown as cartoon and L$_{142}$ as stick representation

positons of the β-hairpin were only marginally rearranged during the docking procedure. The hydrophobic pocket of Vps29 accommodates the TBC1D5 peptide, as well as the RidL β-hairpin (Fig. 4b), showing that binding of the two Vps29 ligands is mutually exclusive (Fig. 4c). Because the RidL β-hairpin is comparatively stiff due to the presence of three prolines, the interaction with Vps29 is likely established through a "knob-into-hole" binding mechanism, a notion supported by

the very fast association rates observed by SPR. The interaction places the N-terminal domain of RidL such that it does not sterically interfere with Vps35, which binds at the opposite side of Vps29 (Fig. 4a). In summary, information-driven docking of RidL and Vps29 revealed that the N-terminal fragment of the effector snuggly fits into the hydrophobic pocket of the retromer subunit.

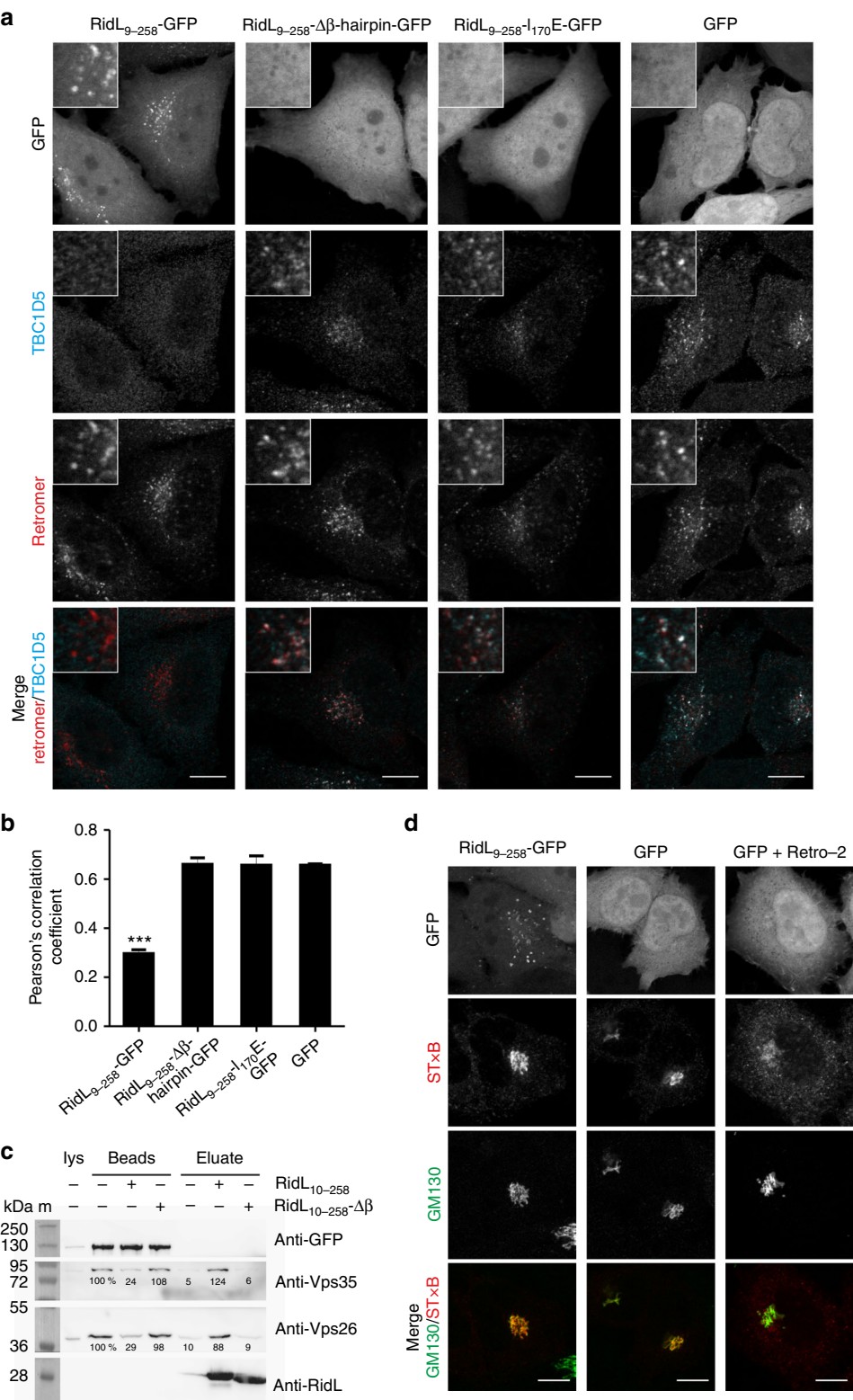

**The RidL$_{9-258}$ β-hairpin displaces TBC1D5 from the retromer**.
Given that RidL and TBC1D5 interact with Vps29 through the
same hydrophobic pocket in the retromer subunit, we tested
whether the *L. pneumophila* effector can displace the regulator
from the retromer. To this end, the RidL fragments RidL$_{9-258}$-
GFP, RidL$_{9-258}$-Δβ-hairpin-GFP, or RidL$_{9-258}$-I$_{170}$E-GFP were
constitutively produced in HeLa cells, and co-localization of
TBC1D5 with the retromer was assessed by using antibodies
recognizing the Vps26 retromer subunit or TBC1D5, respectively
(Fig. 5a).

Upon overproduction of RidL$_{9-258}$-GFP, TBC1D5 was dis-
placed from the retromer, and instead, the RidL fragment co-
localized with the complex. Pearson's correlation coefficient of
TBC1D5 and the retromer was approximately 0.3 upon
production of RidL$_{9-258}$-GFP and 0.7 upon production of GFP,
respectively (Fig. 5b), indicating that the co-localization of
TBC1D5 with the retromer was significantly reduced in the
presence of the N-terminal fragment of RidL. In contrast, in cells
producing RidL$_{9-258}$-Δβ-hairpin-GFP or RidL$_{9-258}$-I$_{170}$E-GFP,
TBC1D5 still co-localized with the retromer, and the GFP-
fusion proteins, as well as GFP, were found throughout the cell.
The GFP fusions of RidL$_{9-258}$, RidL$_{9-258}$-Δβ-hairpin, and RidL$_{9-258}$-I$_{170}$E were produced in HeLa cells in similar amounts, ruling
out protein concentration effects on the displacement of TBC1D5
(Supplementary Fig. 2). To confirm in vitro that the N-terminal
domain of RidL displaces TBC1D5 from the retromer, we
precipitated GFP–TBC1D5 and the associated retromer from
transiently transfected HeLa cells ectopically producing
GFP–TBC1D5. Upon addition of purified RidL$_{10-258}$, the retro-
mer (as detected by Vps26 and Vps35) eluted from the beads
(Fig. 5c and Supplementary Fig. 3). In contrast, the addition of
RidL$_{10-258}$-Δβ-hairpin or buffer only did not result in the elution
of the retromer. Thus, RidL$_{10-258}$ displaces the retromer from
TBC1D5 in a β-hairpin-dependent manner.

Next, we assessed whether the displacement of TBC1D5 by
RidL$_{9-258}$-GFP affects retrograde trafficking of fluorescently
labeled Shiga toxin subunit B (STxB) (Fig. 5d). HeLa cells were
transiently transfected with vectors producing RidL$_{9-258}$-GFP or
GFP, and the trafficking of STxB-Cy3 was analyzed by co-
localization with the Golgi marker GM130. Under these
conditions, the inhibitor Retro-2 abolished retrograde trafficking
of STxB-Cy3, while RidL$_{9-258}$-GFP had no effect. Similarly, upon
ectopic production of RidL$_{9-258}$ or RidL$_{9-258}$-Δβ-hairpin in *D.
discoideum*, the intracellular replication of *L. pneumophila* JR32
or Δ*ridL* was not affected (Supplementary Fig. 5). Taken together,
the retromer-binding RidL fragment RidL$_{9-258}$ displaces TBC1D5
from the retromer in a β-hairpin-dependent manner, but does not
affect retrograde trafficking of STxB or intracellular replication of
the pathogen in *D. discoideum*.

**TBC1D5 localizes to LCVs and promotes *L. pneumophila*
growth**. The small GTPase Rab7 localizes to LCVs in mammalian
cells as well as in *D. discoideum* amoebae[7,8,44], and mammalian
TBC1D5 functions as a Rab7 GAP[25]. To test whether and when
TBC1D5 localizes to LCVs, we quantitatively analyzed by imaging
flow cytometry (IFC) LCVs from *L. pneumophila*-infected *D.
discoideum* producing in tandem GFP–TBC1D5 and the endo-
somal marker AmtA-mCherry (Fig. 6a). The amoebae were
infected with mPlum-producing *L. pneumophila* strain JR32 or
Δ*icmT* mutant bacteria, homogenized, and IFC images of >3000
AmtA-positive cells each at three different time points were
assessed. The TBC1D5 intensity increased on AmtA-positive
vacuoles harboring the parental strain JR32, but not the Δ*icmT*
mutant, and the signal peaked at 2 h post infection (p.i.) (Fig. 6b).

In order to assess whether translocated RidL affects the
localization of TBC1D5 to the pathogen vacuole, we used *D.
discoideum* stably producing GFP-TBC1D5. The amoebae were
infected with DsRed-producing *L. pneumophila* JR32 or Δ*ridL*
harboring pCR77 (vector), Δ*ridL*/pIF007 (RidL), or Δ*ridL*/
pKB201 (RidL-Δβ-hairpin). Intact LCVs were purified,
immuno-stained for calnexin, and analyzed by fluorescence
microscopy (Fig. 6c). LCVs harboring *L. pneumophila* lacking
RidL (Δ*ridL*) or producing only full-length RidL-Δβ-hairpin (not
binding to Vps29; Figs. 2 and 5), were decorated with
approximately 20% more TBC1D5 than LCVs harboring strains
producing RidL (Fig. 6d). At the same time, the amount of
calnexin on pathogen vacuoles harboring these strains was
identical (Supplementary Fig. 6). The mutant protein RidL-Δβ-
hairpin was translocated by *L. pneumophila* into *D. discoideum*
like the wild-type effector protein, ruling out that a translocation
defect of the RidL mutant accounts for the observation (Fig. 6e).
Taken together, these findings are in agreement with the notion
that upon translocation into host cells, RidL displaces TBC1D5
from LCVs by binding to Vps29.

In a similar approach, we infected RAW 264.7 macrophages with
the above-mentioned, DsRed-producing *L. pneumophila* strains
JR32, Δ*ridL*, Δ*ridL*/RidL, or Δ*ridL*/RidL-Δβ-hairpin. To detect
TBC1D5 on LCVs, intact pathogen vacuoles were purified, immuno-
stained for TBC1D5 and SidC, and analyzed. Immunofluorescence
microscopy revealed that also in macrophages, TBC1D5 localizes to
vacuoles harboring these *L. pneumophila* strains (Fig. 6f).

Finally, we tested whether TBC1D5 plays a functional role
during infection with *L. pneumophila*. To this end, we infected *D.
discoideum* DH1 or an isogenic TBC1D5 knockout strain with
virulent *L. pneumophila* JR32 or the avirulent Δ*icmT* strain
producing GFP (Fig. 6g). Upon deletion of TBC1D5, the
intracellular replication of strain JR32 was significantly reduced,
while there was no effect on Δ*icmT*. Concomitantly, deletion of
TBC1D5 had no effect on the biogenesis of the LCV assessed by

**Fig. 5** The RidL$_{9-258}$ β-hairpin displaces TBC1D5 from the retromer. **a** HeLa cells were transiently transfected with vectors producing the GFP-fusion
proteins or GFP as indicated and immuno-stained for TBC1D5 (cyan) and the retromer (Vps26; red). Scale bar, 10 μm. **b** Co-localization of TBC1D5 with
retromer (Vps26) shown in **a** was scored. Pearson's correlation coefficient was calculated in three independent experiments with at least 50 cells each
(mean and SD of the means from each experiment; one way ANOVA; ***$P < 0.001$). **c** GFP-TBC1D5 and associated endogenous retromer components
were pulled down from transiently transfected HeLa cell lysate. The retromer was eluted from immobilized GFP-TBC1D5 with purified RidL$_{10-258}$,
RidL$_{10-258}$-Δβ-hairpin or no protein. Input lysate (lys, 1%), washed beads after elution (beads, 20%) and eluates (20%) were analyzed by SDS-PAGE,
Western blot and immuno-staining of GFP (GFP-TBC1D5: 117 kDa), Vps35 (91 kDa), Vps26 (40 kDa) or RidL (RidL$_{10-258}$: 29 kDa, RidL$_{10-258}$-Δβ-hairpin:
27 kDa). The bands of retromer components were quantified relatively to the value for mock-treated beads (indicated in %). One representative
experiment of three independent biological replicates is shown. m, marker. **d** HeLa cells were transiently transfected with vectors producing RidL$_{9-258}$-GFP
or GFP, and the trafficking of Shiga toxin subunit B (STxB)-Cy3 was analyzed by co-localization with the Golgi-marker GM130 after 30 min. RidL$_{9-258}$-GFP
did not affect trafficking of STxB-Cy3. As a positive control, GFP-producing cells were treated with 25 μM Retro-2 30 min prior to and during the trafficking
assay. Images shown are representative of three independent experiments. Scale bar, 10 μm

measuring the recruitment of calnexin to the LCV over time (Supplementary Fig. 6). In summary, TBC1D5 preferentially localizes to LCVs harboring *L. pneumophila* lacking functional RidL, and TBC1D5 positively affects the intracellular growth of *L. pneumophila*.

**D. discoideum TBC1D5 reduces Rab7 on LCVs**. To test whether *D. discoideum* TBC1D5 affects Rab7 on LCVs, we quantitatively analyzed by IFC the presence of GFP-Rab7 on LCVs in *D.*

*discoideum* DH1 or an isogenic *TBC1D5* knockout strain (Fig. 7a). The amoebae were infected with mPlum-producing *L. pneumophila* JR32, and IFC images of 10,000 cells each at 3 different time points were assessed. The IFC co-localization score (see Materials and Methods section) for the acquisition of GFP-Rab7 to LCVs increased over time, and was 30% higher after 2 h in amoebae lacking TBC1D5 (Fig. 7b). Under the same conditions, the absence of RidL had no effect on Rab7 on LCVs in *D. discoideum* DH1, similar to what has been described in other *D. discoideum* strains[18], and vacuoles harboring $\Delta icmT$ did not

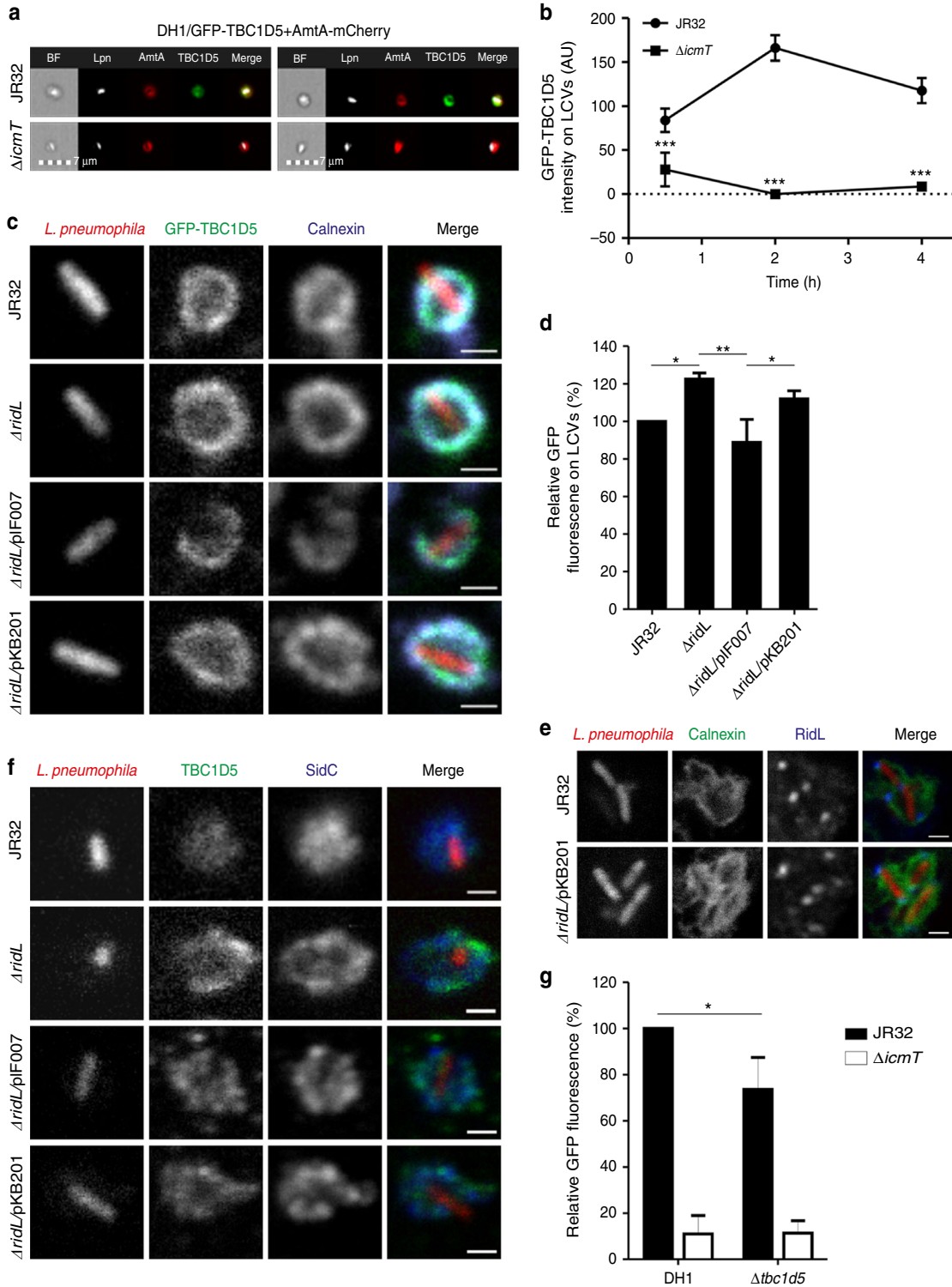

acquire the small GTPase (Fig. 7c, d). In summary, these findings are in agreement with a function of *D. discoideum* TBC1D5 as a Rab7 GAP, which is active on LCVs.

## Discussion

The 29-kDa N-terminal domain of RidL adopts a foot-like fold comprising a hydrophobic β-hairpin at its heel. Through this N-terminal fragment, RidL binds to the retromer subunit Vps29, and the deletion of the β-hairpin or the replacement by glutamate of the hydrophobic residues Ile$_{170}$ in the RidL β-hairpin or Leu$_{152}$ in Vps29 abolishes the RidL-Vps29 interaction in eukaryotic cells and in vitro.

The β-hairpin in RidL$_{2-281}$ protrudes from the protein and adopts a rigid conformation (Fig. 1b). The B-factors of the β-hairpin are comparatively low, as compared with, e.g., the flexible F-loop. This suggests that the β-hairpin is solidly attached to the remainder of the N-terminal domain. Moreover, the structure shows that there are multiple molecular interactions between the β-hairpin and the rest of RidL. This architecture provides a rationale for the observation that the β-hairpin-dependent binding of RidL into the hydrophobic pocket of Vps29 proceeds at a very fast ($> 10^6\,M^{-1}\,s^{-1}$) association rate (Fig. 3 and Supplementary Fig. 4). Hence, the association between Vps29 and RidL is diffusion-limited and very likely does not involve conformational rearrangements of the binding partners[45]. This is in agreement with a "knob-into-hole" type of interaction, rather than a much slower "induced fit" binding mechanism. Of note, the lack of major conformational rearrangements during complex formation was an important prerequisite to predict the structural basis of the Vps29-RidL interface by information-driven docking.

Experimental results and docking studies indicated that the RidL N-terminal β-hairpin interacts with the site on Vps29, where the endogenous mammalian regulator TBC1D5 also binds (Figs. 2, 4). Accordingly, binding of the N-terminal RidL fragment to Vps29 displaced TBC1D5 from the retromer in cells and in vitro (Fig. 5), and RidL reduced TBC1D5 on LCVs in *L. pneumophila*-infected *D. discoideum* (Fig. 6). Displacement of TBC1D5 from the retromer might liberate the regulator to function in pathways other than the retrograde route. Accordingly, TBC1D5 might be involved in the late endocytic or autophagy pathways by targeting Rab7 or Atg9, respectively[46]. In this scenario, TBC1D5 would inhibit cellular-trafficking pathways restricting *L. pneumophila*, and thus promote the intracellular growth of the pathogen. In agreement with a beneficial effect of TBC1D5 for intracellular replication of *L. pneumophila*, the deletion of the regulator in *D. discoideum* reduced the growth of

the pathogen (Fig. 6). The deletion of *D. discoideum* TBC1D5 increased Rab7 on LCVs (Fig. 7), suggesting that its putative GAP activity indeed inactivates Rab7 similar to the mammalian Rab7 GAP TBC1D5[25].

Rab7 is a pivotal regulator of retrograde trafficking in mammalian cells[24], as well as in the protozoon *Entamoeba histolytica*[47]. The active, GTP-bound form of Rab7 interacts with the retromer. Depletion of Rab7 or a dominant-negative form of the GTPase caused the retromer (but not SNX dimers) to dissociate from endosomes[24]. Rab7 localizes to LCVs in *D. discoideum* as well as in mammalian cells[7,8,44]. However, the depletion of Rab7 does not significantly affect the intracellular replication of *L. pneumophila*, at least in mammalian cells[8]. While Rab7 and TBC1D5 control the recruitment and release of the retromer to endosomes[24–26], *L. pneumophila* RidL does not appear to affect the decoration of LCVs with Rab7 (Fig. 7) or the retromer in *D. discoideum* or macrophages[18], and the N-terminal RidL$_{9-258}$ fragment is not sufficient to inhibit retrograde trafficking (Fig. 5) or promote intracellular replication (Supplementary Fig. 5). Therefore, *D. discoideum* TBC1D5 might play a role during *L. pneumophila* infection beyond the inactivation of Rab7 on LCVs or target other Rab GTPases.

Other than Rab7, a number of small GTPases indeed affects the intracellular replication of *L. pneumophila*. Depletion by RNA interference revealed that endocytic GTPases such as Rab5, Rab14, and Rab21 restrict the intracellular growth of *L. pneumophila*, whereas secretory GTPases such as Rab8, Rab10, and Rab32 implicated in Golgi to endosome trafficking promote the replication of the pathogen[8]. These small GTPases localize to vacuoles harboring virulent *L. pneumophila* JR32 but not Δ*icmT* mutant bacteria[7,8]. The finding that depletion of endosomal small GTPases promotes *L. pneumophila* replication suggests that a corresponding GAP would inversely affect the pathogen, i.e., its depletion or deletion would impair bacterial growth. Indeed, the deletion of *D. discoideum* TBC1D5 restricts the intracellular replication of *L. pneumophila* (Fig. 6). This result is in agreement with a potential GAP activity of TBC1D5 towards Rab5, Rab14, and/or Rab21, and perhaps, TBC1D5 shows a relaxed GAP activity, which is relevant in the context of an *L. pneumophila* infection. Alternatively, a rather specific Rab7 GAP activity of TBC1D5 might indirectly affect the activity of another GTPase that is more relevant for *L. pneumophila* infection than Rab7. Given that TBC1D5 appears to play a role for the release of the retromer from Rab7 at later steps in retrograde trafficking[25], RidL might also target later steps in the process. However, the exact role of the retromer interactor TBC1D5 for *L. pneumophila* infection requires further studies.

**Fig. 6** TBC1D5 localizes to LCVs and promotes growth of *L. pneumophila*. **a** Imaging flow cytometry (IFC) images of LCVs from homogenized *D. discoideum* DH1 producing in tandem GFP-TBC1D5 (pFL1304) and AmtA-mCherry as an LCV marker, infected (MOI 50, 2 h) with mPlum-producing *L. pneumophila* JR32 or Δ*icmT* (pAW14). **b** Quantification by IFC of the intensity of GFP-TBC1D5 on >3000 LCVs per sample at the time points post-infection indicated. The data show mean and 95% confidence intervals of one representative experiment out of three independent biological replicates (***$P < 0.001$; two-way ANOVA with Bonferroni post hoc test). **c** *D. discoideum* DH1 stably producing GFP-TBC1D5 was infected (MOI 20, 2 h) with DsRed-producing *L. pneumophila* JR32 or Δ*ridL* harboring pCR77 (vector), Δ*ridL*/pIF007 (RidL) or Δ*ridL*/pKB201 (RidL-Δβ-hairpin). Intact LCVs were purified, immuno-stained for calnexin, and analyzed by fluorescence microscopy. Scale bar, 1 μm. **d** Quantification of **c**. Mean and SD of the means of three independent biological replicates (LCV isolations) are shown, each with at least 44 LCVs (average pixel GFP intensity of masked LCVs relative to the value of strain JR32; one way ANOVA; *$P < 0.05$, **$P < 0.01$). **e** *D. discoideum* Ax3 stably producing calnexin-GFP were infected (MOI 10, 1 h) with DsRed-producing *L. pneumophila* JR32 harboring pCR77 (vector), or Δ*ridL*/pKB201 (RidL-Δβ-hairpin), immuno-stained for RidL, and effector translocation was analyzed by fluorescence microscopy. Scale bar 1 μm. **f** RAW 264.7 macrophages were infected (MOI 10, 2 h) with DsRed-producing *L. pneumophila* JR32 or Δ*ridL* harboring pCR77 (vector), Δ*ridL*/pIF007 (RidL) or Δ*ridL*/pKB201 (RidL-Δβ-hairpin). Intact LCVs were purified, immuno-stained for TBC1D5 and SidC, and analyzed by fluorescence microscopy. Scale bar, 1 μm. **g** *D. discoideum* DH1 or an isogenic TBC1D5 knockout strain (Δ*tbc1d5*) was infected (MOI 10, 24 h) with GFP-producing *L. pneumophila* JR32 (black bars) or Δ*icmT* mutant bacteria (white bars) harboring plasmid pNT-28, and intracellular replication was determined by fluorescence. Mean of fluorescence relative to JR32 in DH1 and SD of four independent experiments (each in technical triplicates) are shown (paired Student's *t* test; *$P < 0.05$)

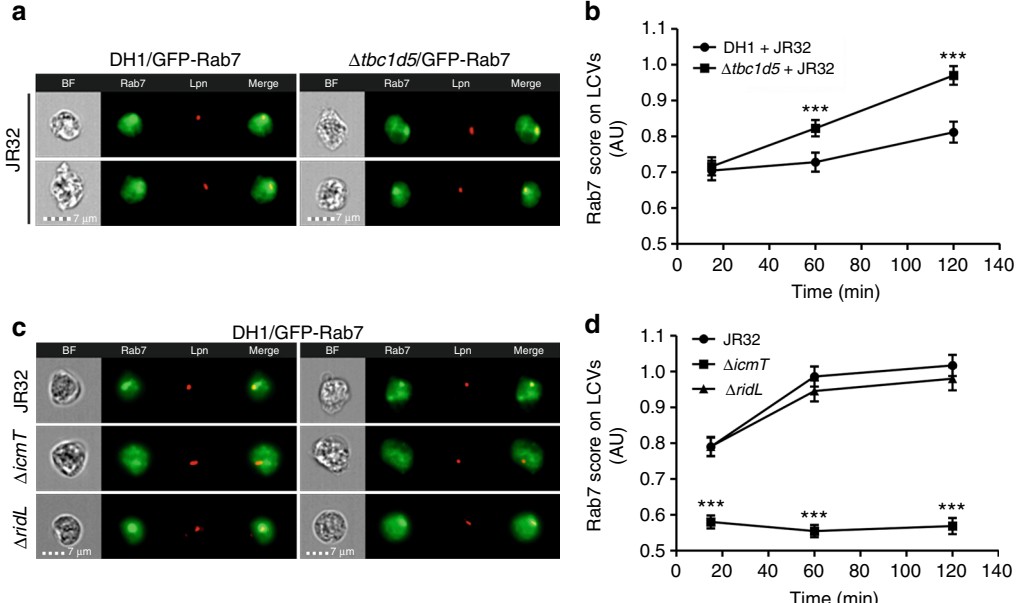

**Fig. 7** *D. discoideum* TBC1D5 reduces Rab7 on LCVs. **a** Imaging flow cytometry (IFC) images of *D. discoideum* DH1 (WT or Δ*tbc1d5*) producing GFP-Rab7 (pAW9), infected (MOI 5, 2 h) with mPlum-producing *L. pneumophila* JR32 (pAW14). **b** Quantification of IFC co-localization score between GFP and mPlum in >750 cells per sample at the time points p.i. indicated. The data show mean and 95% confidence intervals of one representative experiment out of four independent experiments (\*\*\**P* < 0.001; two-way ANOVA with Bonferroni post hoc test). **c** IFC images of *D. discoideum* DH1 producing GFP-Rab7 (pAW9), infected (MOI 5, 2 h) with DsRed-producing *L. pneumophila* JR32, Δ*icmT* or Δ*ridL* harboring pCR77. **d** Quantification of IFC co-localization score between GFP and DsRed in >1000 cells per sample at the time points p.i. indicated. The data show mean and 95% confidence intervals of one representative experiment out of four independent experiments (\*\*\**P* < 0.001; two-way ANOVA with Bonferroni post hoc test)

Upon translocation, RidL displaces TBC1D5 from LCVs (Fig. 6); yet, the effect is rather weak. At this point, we cannot exclude that among the ~ 300 *L. pneumophila* effector proteins, other toxins than RidL target the retromer or the retromer-TBC1D5 interaction, which might compromise the effect(s) of RidL. Moreover, while the 29-kDa N-terminal RidL fragment is sufficient to displace TBC1D5, the remaining 102-kDa fragment of RidL very likely harbors additional (perhaps catalytic) activities. Indeed, a number of *L. pneumophila* effector proteins are large (>80 kDa) and represent multidomain proteins with several catalytic activities[31,32,34]. Thus, in addition to anchoring to Vps29 and displacing TBC1D5, RidL likely has other, unknown functions contributing to the inhibition of retrograde trafficking.

The finding that the deletion of TBC1D5 actually reduces intracellular replication of *L. pneumophila* is rather counter-intuitive. Previously, we made another seemingly paradox observation that deletion of OCRL/Dd5P4 (a PtdIns(4,5)$P_2$ 5-phosphatase producing PtdIns(4)$P$ on LCVs, which *L. pneumophila* effectors bind to) indeed promotes (rather than reduces) intracellular growth of *L. pneumophila*[17]. TBC1D5 as well as OCRL likely exhibit pleiotropic and complex functions in eukaryotic cells, which collectively define the (positive or negative) role adopted by the host factors during *L. pneumophila* infection.

Our current working model of the mode of action of RidL includes the competition of the effector with the retromer inter-actor TBC1D5 and its displacement from the Vps29 subunit of the coat complex (Fig. 8). To initiate retrograde trafficking, the retromer (Vps29-Vps35-Vps26) and TBC1D5 are recruited to the membrane by activated GTP-bound Rab7, and cargo is recruited by the retromer. The retromer interacts with membrane-deforming sorting nexin (SNX) dimers, and the Rab7 GAP TBC1D5 inactivates and releases Rab7 in order for retrograde trafficking to proceed along microtubules. In the presence of *L. pneumophila* RidL, TBC1D5 is displaced from Vps29, and retrograde trafficking is blocked by an unknown mechanism. These findings pave the way for further studies, addressing the molecular mechanism of retrograde-trafficking inhibition by RidL.

## Methods

**Bacteria and cell infection**. Bacterial strains used in this study are listed in Supplementary Table 1. *L. pneumophila* strains were grown for 3 days on CYE agar plates containing charcoal yeast extract, buffered with N-(2-acetamido)-2-amino-ethanesulfonic acid (ACES). Liquid cultures were inoculated in AYE medium at an OD$_{600}$ of 0.1–0.2 and grown at 37 °C to an OD$_{600}$ of 5.0 (21–22 h). Chlor-amphenicol (Cam; 5 µg ml$^{-1}$) and isopropyl-1-thio-β-D-galactopyranoside (IPTG; 1 mM) were added if required.

Human HeLa epithelial cells (lab collection[40]) and murine macrophage-like RAW 264.7 cells (lab collection[18]) were cultivated in RPMI 1640 medium supplemented with 10% heat-inactivated fetal bovine serum (FBS) and 1% glutamine (all from Life Technologies). The cells were incubated at 37 °C/5% CO$_2$ in a humidified atmosphere.

*Dictyostelium discoideum* strains (Supplementary Table 1) were grown axenically in HL5 medium (ForMedium) at 23 °C and transfected by electroporation as described[48,49]. Electroporation was performed with a Gene Pulser Xcell (BioRad) device using the following conditions: pre-programmed – fungi – *D. discoideum* at 0.85 kV, 3 µF, time constant 1–1.2 and two pulses with an interval of 5 s. After electroporation of plasmids, transfectants were selected and maintained in 10–20 µg ml$^{-1}$ geneticin (G418). The *D. discoideum* TBC1D5 knockout strain was generated as follows. To obtain the knockout vector, the 5′ fragment was amplified from genomic DNA with sense (5′-GTCGACATGGA GGATAATGATATTAGTTTTAGT-3′) and antisense (5′-AAGCTTTTATTC AACCCAAGTGGTTGGATCTTTTGA-3′) oligonucleotides and cloned into pBlueScript vector (Stratagene, La Jolla, CA, USA). The 3′ fragment was obtained by PCR using sense (5′-AGTGTCTCTAGAAAGGATGGAGGTATTGGTGCT TTT-3′) and antisense (5′-GCGGGCCGCTTAATCAACAGATTCAAAGTTA TCTTGATCATG-3′) oligonucleotides and cloned into pBlueScript containing the 5′ fragment. After sequencing, the knockout vector was completed by inserting the blasticidin resistance cassette between the two 5′ and 3′ fragments. The resulting plasmid was linearized (KpnI and NotI digestion) and electroporated into DH1 cells. Transformants were selected in presence of 10 µg ml$^{-1}$ blasticidin. Individual colonies were tested by PCR to confirm *TBC1D5* gene replacement.

For intracellular replication assays, the infection of *D. discoideum* amoebae with *L. pneumophila* producing GFP (pNT-28) was analyzed as described[36,49,50]. Briefly, the amoebae were infected with *L. pneumophila* grown for 21–22 h in AYE broth

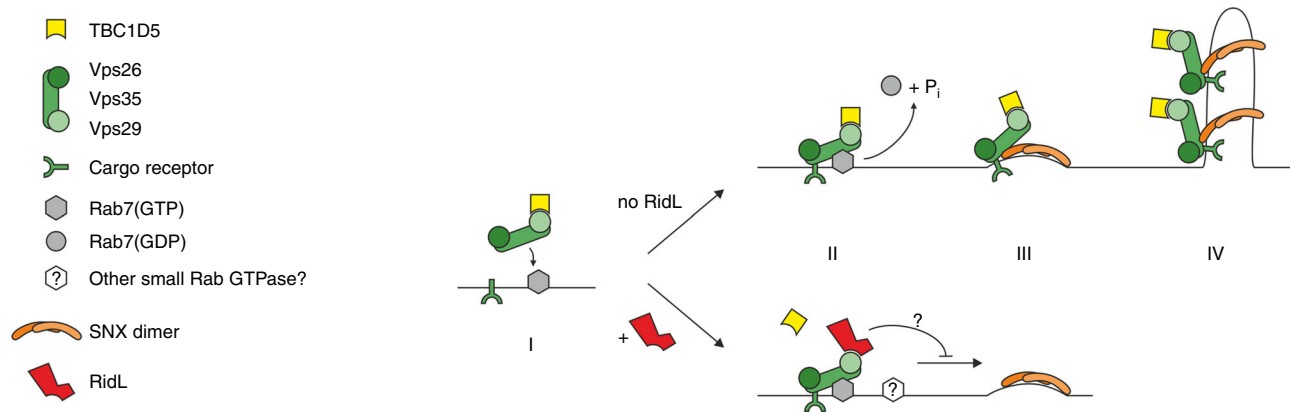

**Fig. 8** Model of RidL displacement of TBC1D5. RidL$_{9-258}$ binds to Vps29 and displaces TBC1D5 from the retromer. (I) Retromer (Vps29-Vps35-Vps26) and TBC1D5 are recruited to the membrane by activated GTP-bound Rab7; cargo is recruited by the retromer. (II) In absence of RidL TBC1D5 leads to GTP hydrolysis and release of Rab7, whereas in presence of RidL TBC1D5 is displaced from Vps29, and retrograde trafficking is blocked by an unknown mechanism. (III) Retromer interacts with membrane-deforming sorting nexin (SNX) dimers. (IV) Transport tubules are formed involving other factors not shown here for simplicity

with the indicated MOI, the infection was synchronized by centrifugation ($450 \times g$, 10 min, RT), the infected amoebae were incubated at 25 °C for the indicated duration, and intracellular replication was assessed by GFP fluorescence with a plate reader (BioTek).

For fluorescence microscopy, *D. discoideum* producing calnexin-GFP was infected as described[17,18,36,40,48]. Briefly, exponentially growing amoebae were seeded on sterile poly-L-lysine-coated coverslips in 24-well plates at $2.5 \times 10^5$ per well in 0.5 ml HL5 medium and let grow overnight. *L. pneumophila* cultures grown to an $OD_{600}$ of 5.0 in AYE medium were diluted in HL5 medium to infect the amoebae by centrifugation (MOI 5), and the infected cells were incubated at 25 °C and fixed with 4% paraformaldehyde (PFA; 30 min, 4 °C) at the time points indicated. Alternatively, *D. discoideum* producing GFP-TBC1D5 was infected in T75 flasks with the indicated *L. pneumophila* strains for 2 h (MOI 20). Intact LCVs were purified by the immuno-affinity two-step procedure as described[51,52]. Briefly, infected amoebae were lysed by 7–9 passages through a stainless steel ball homogenizer (Isobiotec), incubated with an antibody recognizing the LCV-bound effector SidC and a secondary antibody coupled to magnetic beads, enriched with a MACS separator (Miltenyi Biotec), and finally purified by density gradient centrifugation through a Histodenz cushion. Isolated LCVs were spun on poly-L-lysine coated coverslips and stained with an antibody recognizing calnexin (1:200, DSHB Iowa: 270-390-2) and a corresponding secondary antibody (Cy5, 1:250, Jackson: 115-175-044). The samples were viewed with a Leica SP5 Mid UV-VIS confocal microscope (HCX PL APO Leica objective 63×/1.4 oil; Leica Microsystems). For quantification single LCVs were masked, and the average pixel intensity for indicated channels was calculated using ImageJ.

Analogously, RAW 264.7 cells were infected (MOI 10), intact LCVs were purified as described above and stained with antibodies against TBC1D5 (1:50, mouse, Santa Cruz: sc-376296), SidC (1:1000, rabbit[36],) and corresponding secondary anti-mouse (FITC, 1:250, Jackson: 115-095-072) and anti-rabbit (Cy5, 1:200, Invitrogen: A-10523) antibodies.

**Imaging flow cytometry.** For imaging flow cytometry (IFC) with intact amoebae, *D. discoideum* DH1 or a $\Delta TBC1D5$ isogenic mutant amoebae producing GFP-Rab7 (pAW09) were seeded in 12-well plates and infected (MOI 5) with the indicated *L. pneumophila* JR32, $\Delta icmT$ or $\Delta ridL$ strains producing mPlum (pAW14) or DsRed (pCR77). At the indicated time points, the infected amoebae were detached from the surface and fixed with 4% PFA 30 min at 4 °C. With an imaging flow cytometer (ImageStreamX MkII, Amnis), 10,000 events were acquired, and after color compensation, analysis was carried out using the IDEAS 6.2 software (Amnis) as described[9,10,53], by first selecting in-focus single cells that were positive for GFP, and then gating cells having internalized *L. pneumophila* using the IDEAS internalization wizard. The cells containing more than one bacterium were subsequently eliminated using the feature [Spot Count_Spot(M04 or M05, Lpn, Bright, 8.5, 1, 0)_4]. The included cells were finally analyzed for co-localization between mPlum or DsRed and GFP (IFC co-localization score), using the co-localization wizard in IDEAS, which computes the log transformed Pearson's correlation coefficient of the localized bright spots with a radius of 3 pixels or less in two images, thus providing an IFC co-localization score for each cell. The mean score for the sample is reported. The IFC data were analyzed by regular two-way ANOVA followed by Bonferroni post hoc tests, comparing each sample to the corresponding wild-type.

For IFC of isolated LCVs, *D. discoideum* DH1 dually expressing GFP-TBC1D5 (pFL1304) and AmtA-mCherry were seeded in T25 cell culture flasks and infected (MOI 50) with *L. pneumophila* JR32 or $\Delta icmT$ producing mPlum (pAW14). At the indicated time points, the amoebae were detached from the surface, washed, homogenized using a ball homogenizer, and fixed in suspension as described above. With an imaging flow cytometer (ImageStreamX MkII, Amnis), 20,000 mPlum- and AmtA-positive events were acquired, and after color compensation, analysis was carried out using the IDEAS 6.2 software (Amnis). First, in-focus events were gated using the feature [Gradient RMS_M05_Lpn]. Second, intact cells and large membrane aggregates were excluded using the feature [Area_MC]. Third, events displaying a sufficiently high mPlum signal were gated using [Max Pixel_MC_Lpn]. Fourth, LCVs containing one bacterium only were selected using the features (Spot count_Spot(M05, Lpn, Bright, 6, 1, 0)_4) vs. (Area_Object(M05, Lpn, Tight)). Fifth, AmtA-mCherry-positive events with the AmtA signal colocalizing with the mPlum signal were gated using the features (Max Pixel_MC_AmtA) vs. (Bright Detail Similarity R3_MC_Lpn_AmtA). Finally, the GFP intensity on the AmtA-positive LCVs was quantified using (Intensity_MC_GFP), and the mean value for the sample is reported.

**Ectopic production of RidL fragments.** For ectopic production of RidL fragments fused to GFP, Hela cells were transiently transfected with the corresponding plasmids (Supplementary Table 1) in six-well plates using lipofectamine 2000 (Invitrogen: 11668019) according to the manufacturer's protocol. Twenty-four hours after transfection, cells were seeded on coverslips, and after 24 h, cells were fixed with 4% paraformaldehyde (15 min, RT), and permeabilized using 0.1% Triton in PBS (10 min, RT). The cells were immuno-stained with antibodies against Vps26 (1:1000, rabbit, Abcam: ab23892), TBC1D5 (1:50, mouse, Santa Cruz: sc-376296) and corresponding secondary anti-rabbit (Cy5, 1:200, Invitrogen: A-10523) or anti-mouse (Cy3, 1:200, Jackson: 115-167-003) antibodies. The samples were viewed with a Leica SP5 resonant APD confocal microscope (HCX PL APO Leica objective 63x/1.4 oil; Leica Microsystems). For quantification, single cells were masked, and Pearson's correlation coefficients between the indicated channels were calculated using ImageJ (Image pixel size: 120 nm).

To control ectopic production of GFP-fusion proteins in transfected HeLa cells, the cells were detached after 24 h and lysed in 2× loading buffer (95 °C, 15 min). Total cell extract was separated on 10% SDS-polyacrylamide gels, blotted onto nitrocellulose and stained for either GFP (1:5000, mouse, Clontech: 632380) or RidL (1:1000, rabbit[18],), followed by corresponding secondary antibodies linked to HRPO (anti-mouse, 1:5000, GE Healthcare: NA931, or anti-rabbit, 1:2000, GE Healthcare: NA934) and detection by chemiluminescence (Amersham ECL kit; GE Healthcare: RPN2109).

To measure retrograde trafficking of Shiga toxin subunit B (STxB)-Cy3 (gift of Ludger Johannes), HeLa cells were transfected as described above and seeded on coverslips. The cells were incubated on ice for 30 min in presence of 0.5 µg ml$^{-1}$ STxB-Cy3, washed and incubated in new medium for 30 min at 37 °C. The cells were fixed and stained for GM130 (1:200, mouse, BD biosciences: 610823) followed by secondary anti-mouse antibody (Cy5, 1:250, Jackson: 315-175-044). As a positive control cells were treated with 25 µM Retro-2 (in DMSO, Sigma-Aldrich: SML1085) 30 min prior to and during all incubation steps.

**Pull-down and in vitro TBC1D5 displacement.** For in vitro displacement of TBC1D5 from retromer by RidL, HeLa cells were transiently transfected with peGFP-C1-TBC1D5 using lipofectamine 3000 (Invitrogen: L3000008) according to the manufacturer's protocol. Twenty-four hours after transfection, the cells were transferred to T75 flasks and incubated for 48 h. The cells were trypsinized, washed in PBS and lysed in PBS containing 0.2% NP-40 and protease inhibitor (Roche: cOmplete tablet). GFP-tagged TBC1D5 and associated retromer components were

pulled down using GFP-trap (Chromotek: gta-20) according to the manufacturer's protocol. Briefly, 60 μl beads were incubated with cell lysate supernatant, washed five times with lysis buffer, split into three samples and incubated for 30 min with 15 μg purified RidL$_{10-258}$, RidL$_{10-258}$-Δβ-hairpin or buffer only, respectively. Eluates were collected and beads were washed four times with lysis buffer before resuspending in loading buffer. 20% of total eluate or bead suspension, respectively, were analyzed by SDS-PAGE and Western blot as described above. The membrane was cut according to the expected target sizes and different pieces were individually immuno-stained for GFP or RidL as described above, or for Vps35 (1:1000, Abcam: ab10099) or Vps26 (1:1000, Abcam: ab23892). The bands were quantified using the ImageQuant QL software relative to the value for beads with mock elution.

**Molecular cloning.** Plasmids and oligonucleotides are listed in Supplementary Tables 1 and 2, respectively. DNA manipulations were performed according to standard protocols, and plasmids were isolated using kits from Qiagen (EndoFree Plasmid Maxi Kit, 12362) for transfection grade plasmids or from Macherey-Nagel. All PCR fragments were sequenced. Translational His-fusion proteins of *ridL* variants were constructed by FX-cloning[54]. Briefly, the corresponding DNA was amplified via PCR from pCR090, pKB151, or pEF-Bos-Flag-Vps29 using the primers oKB001/2, oKB076/82, or oKB003/4, respectively. The fragments were cut with BspQI and inserted into pINIT, yielding the plasmids pKB002, pKB158, or pKB004, respectively. Analogously, amplification from pKB002 with primer pairs oKB001/80, oKB076/82, or oKB098/oKB002 resulted in pKB128, pKB136, or pKB164, respectively. Inserts from pKB128, pKB136, or pKB158 were excised with BspQI and inserted into pBXNH3, yielding the plasmids pKB134, pKB137, or pKB161. Analogously, inserts from pKB002, pKB004, pKB044, pKB164, pKB151, or pKB152 were excised and inserted into pBXC3GH, yielding the plasmids pKB016, pKB024, pKB045, pKB167, pKB153 or pKB154, respectively. Translational Avi-fusions were constructed by excising inserts from pKB002, pKB004, or pKB044 with BspQI and insertion into pBXCA3GH, yielding the plasmids pKB108, pKB109, or pKB144, respectively.

Loop deletion mutants were constructed by site-directed mutagenesis. For the Δβ-hairpin mutant, PCR products amplified from pCR094 with primers oKB088/85 and oKB084/89, respectively, were mixed and amplified again using the primers oKB088/89. The resulting DNA fragment was cut with SalI/ApaI and inserted into pCR094, yielding the plasmid pKB149. From pKB149, a DNA fragment carrying the mutation was transferred into pKB002 using XhoI resulting in the plasmid pKB151. The ΔF-loop mutant was constructed analogously using the primer pairs oKB088/91 and oKB90/89, resulting in the plasmids pKB150 (combined fragments inserted into pCR094) and pKB152 (mutation further transferred into pKB002). The RidL$_{I170E}$ point mutation was constructed by site-directed mutagenesis of the corresponding gene in pKB235, using the primer pair oKB136/137 to amplify the whole vector from pKB002. Analogously, the Vps29$_{L152E}$ point mutation was constructed by mutagenesis of the corresponding gene in pKB044 using the primer pair oKB038/39 and pKB004 as a template.

To construct translational GFP-fusions of RidL variants, DNA was amplified from pKB002 and pKB151 using the primers oKB095/116 and oCR126/oKB132, respectively. The fragments were cut with SalI/BamHI and inserted into peGFP-N1, yielding the plasmids pKB185 and pKB216, respectively. Plasmid pKB234 and pKB236 were constructed by insertion of a Bpu1102I/XagI fragment excised from pKB216 and pKB235, respectively. Similarly, amplification from pKB002 and pKB151 with the primer pair oKB127/oKB130 followed by insertion with BglII/BcuI into pDM323 resulted in plasmids pKB203 and pKB269, respectively.

To obtain the GFP-TBC1D5 fusion construct pFL1304, the *D. discoideum* TBC1D5 coding sequence (DDB_G0280253) was amplified by PCR and cloned into pDXA-3C[55]. The plasmid for overproduction of Rab7 with an N-terminal GFP-tag (pAW9) was constructed by amplifying the *D. discoideum* Rab7A gene from pSU23 with the primers oAW13/oAW14, and cloning the PCR fragment into pDM317 using BamHI/SpeI. To generate the plasmid pAW14[10] for constitutive production of far red fluorescent mPlum, the gene and its ribosome-binding site (RBS) were amplified by PCR from mPlum-pBAD using the primers oAW30_fwd and oAW31_rev, and ligated into pNT28 cut with EcoRI/HindIII. Plasmid pAW16 for production of calnexin with a C-terminal GFP-tag was constructed by amplifying the *D. discoideum cnxA* gene from pWS21 with the primers oAW35/oAW36, and cloning the PCR fragment into pDM323 using BamHI/SpeI.

**Protein purification and crystallization.** For the production of RidL$_{2-281}$ (pKB134) and RidL$_{10-258}$-Δβ-hairpin (pKB161) *E. coli* MC1061 harboring the respective pBXNH3-derivative (N-terminal cleavable His$_{10}$-tag) was induced at a cell density (OD$_{600}$) of 0.5–0.8 with 0.02% (w/v) L(+)-arabinose overnight at 15 °C in LB medium containing 100 μg ml$^{-1}$ ampicillin. Selenomethionine (Se-Met) substituted RidL$_{2-281}$ protein (RidL$_{2-281}$$^{Se-Met}$) was produced using *E. coli* MC1061 harboring pKB134 induced as above in M63 minimal medium supplemented with 0.3% (v/v) glycerol, vitamins, amino acids and Se-Met (50 mg l$^{-1}$).

Bacteria were resuspended in buffer (20 mM TRIS/HCl pH 8, 200 mM NaCl supplemented with 3 mM MgSO$_4$, 1 mM PMSF and DNAse (Sigma)) and lysed by high-pressure homogenization (microfluidizer). The following steps were performed at 4 °C. The His-fusion proteins were purified via Ni$^{2+}$-NTA chromatography (washing buffer: 50 mM imidazole pH 7.5, 200 mM NaCl, 10% glycerol; elution buffer: washing buffer with 200 mM imidazole). In order to

remove imidazole, the buffer was exchanged to size-exclusion chromatography (SEC) buffer (20 mM TRIS/HCl pH 7.4, 150 mM NaCl) using PD-10 desalting columns (GE Healthcare). The His-tag was cleaved overnight with His-tagged 3 C protease (produced in house; 10 μg ml$^{-1}$), and the protein mixture was reloaded on a Ni$^{2+}$-NTA column to remove the protease and cleaved His-tag. Proteins were further purified via SEC (Superdex 200 Increase 10/300 GL; GE Healthcare) using SEC buffer as running buffer. Fractions of the monodisperse peak were pooled and concentrated to 13 mg ml$^{-1}$ (18 mg ml$^{-1}$ for RidL$_{2-281}$) using an Amicon Ultra-4 concentrator unit with a MWCO of 10 kDa (and additionally for RidL$_{2-281}$$^{Se-Met}$ an Amicon Microcon microconcentrator unit with a MWCO of 10 kDa) prior to crystallization. Protein concentration was determined by OD$_{280}$ using a NanoDrop 2000 photospectrometer and calculated based on theoretical extinction coefficients (www.expasy.ch/tools/ protparam.html). Crystals of RidL$_{2-281}$$^{Se-Met}$ were obtained by the vapor diffusion method in sitting drops at 20 °C against a reservoir containing 20% (w/v) polyethylene glycol (PEG) 8000, 100 mM TRIS pH 8.5 and 200 mM MgCl$_2$. Analogously, crystals of RidL$_{10-258}$-Δβ-hairpin were obtained against a reservoir containing 25% (w/v) PEG 1500. The protein to reservoir volume ratio was 1:1. The crystals were cryoprotected in mother liquor supplemented with 20% glycerol and flash-frozen in liquid nitrogen.

**Structure determination.** The diffraction data of the crystals were collected at the beamline X06DA of the Swiss Light Source (SLS, Villigen, Switzerland) equipped with a PILATUS detector (Dectris) and processed using XDS[56]. The structure of RidL$_{2-281}$ was solved by single-wavelength anomalous dispersion (SAD) using data from a selenomethionine derivative crystal collected at a wavelength of 0.97852 Å (peak). The anomalous signal extended to 3.1 Å, and SHELXD was used to determine the three selenomethionine positions. SHELXE was used to determine initial phases and for poly-alanine tracing[57]. The coordinates obtained from SHELXE served as a starting point for PHENIX AutoBuild using the native 1.9 Å data set, which resulted in a reasonable first structure. The structure was finalized manually using COOT[58] for building and PHENIX[59] for refinement. The structure of RidL$_{10-258}$-Δβ-hairpin was solved by molecular replacement using PHASER[60].

**Size-exclusion chromatography and surface plasmon resonance.** Proteins for interaction experiments were purified as described above unless stated otherwise, supplemented with 10% glycerol, flash-frozen in liquid nitrogen and kept at −80 °C until use. Full-length RidL$_{2-1167}$ (pKB016), Vps29 (pKB024), Vps29-L$_{152}$E (pKB045), RidL$_{2-1167}$-Δβ-hairpin (pKB153), RidL$_{2-1167}$-ΔF-loop (pKB154) and RidL$_{259-1167}$ (pKB167) were purified via a cleavable C-terminal His$_{10}$-GFP-tag, RidL$_{10-258}$ (pKB137) via a cleavable N-terminal His$_{10}$-tag, and C-terminal Avi-fusion proteins were purified via a cleavable C-terminal His$_{10}$-GFP-tag (RidL$_{2-1167}$-Avi (pKB108), Vps29-Avi (pKB109) and Vps29-L$_{152}$E-Avi (pKB144)). The enzymatic, site-specific biotinylation of the Avi-tag was carried out in vitro using purified BirA[61].

For interaction measurements by SEC, purified RidL$_{2-1167}$, RidL$_{2-1167}$-Δβ-hairpin or RidL$_{2-1167}$-ΔF-loop were mixed at a molar ratio of 1:1.5 with Vps29 (and Vps29-L$_{152}$E for RidL$_{2-1167}$) and subjected to SEC as described above (Superdex 200 Increase 10/300 GL, GE Healthcare). 0.5 ml elution fractions were collected and analyzed for co-elution of the mixed proteins via SDS-PAGE and Coomassie Brilliant Blue staining.

The affinities of RidL$_{2-1167}$, RidL$_{10-258}$, RidL$_{259-1167}$, RidL$_{2-1167}$-Δβ-hairpin and RidL$_{2-1167}$-ΔF-loop towards biotinylated Vps29(-L$_{152}$E)-Avi were determined by surface plasmon resonance (SPR) on a ProteOn XPR36 machine (BioRad). SEC as the last step of purification and SPR were performed with SEC buffer containing 0.05% Tween-20 (v/v). For the SPR measurement, 300 response units (RU) of biotinylated target protein were immobilized on a NeutrAvidin-coated ProteOn™ NLC Sensor Chip (Biorad). For each RidL variant, a 3-fold dilution series of four different concentrations were used for measurements (50–1.35 μM). Injection time was set to 240 s. In an inverted setup, biotinylated RidL$_{2-1167}$-Avi was immobilized (1000 RU) and Vps29 was used as an analyte (50–1.35 μM). Injection time was 150 sec. Responses at equilibrium were plotted against Vps29 concentration. The equilibrium constant $K_D$ was determined by non-linear regression using an equilibrium binding equation equivalent to the Michaelis-Menten equation in which R$_{eq}$ denotes the SPR response at equilibrium, R$_{max}$ denotes the maximal SPR response and [Vps29] is the Vps29 concentration: R$_{eq}$ = R$_{max}$ [Vps29]/($K_D$ + [Vps29]).

**Docking studies.** Information-driven docking was performed using the HAD-DOCK2.2 webserver (http://milou.science.uu.nl/services/HADDOCK2.2/)[62,63]. The RidL$_{2-281}$ structure (PDB: 5OH5) and the Vps29 structure (PDB: 5GTU, with the peptide TBC1D5-Ins1 removed) were used for docking. The amino acids Met$_{-3}$ to His$_0$ of the Vps29 structure (PDB: 5GTU), which must be cloning artifacts, were removed, since they are located in close proximity to the anticipated binding region. We defined Ile$_{170}$ of RidL and Leu$_{25}$, Leu$_{152}$ and Tyr$_{165}$ of Vps29 as active residues and ran the docking with default settings using the "Easy interface"[59]. In the two best scored solutions (HADDOCK score of −65.9 and −56.6, respectively), the RidL β-hairpin is only partially accommodated by the hydrophobic groove of Vps29, and Ile170 of RidL was not contacting Leu152 of Vps29, while other surface regions of RidL, which were not being part of the β-hairpin, interacted strongly with the surface of Vps29. The third best solution (HADDOCK

score −55.6), on the other hand, featured a contact between Ile[170] of RidL and Leu[152] of Vps29, and RidL interacts with Vps29 exclusively via the β-hairpin. Therefore, we show the results of this highly ranked solution.

**Data availability**. The coordinates of *Legionella pneumophila* RidL N-terminal retromer-binding domain were deposited at the protein database under accession codes 5OH5 and 5OH6 (lacking β-hairpin). The data sets generated and/or analyzed during the current study are available from the corresponding authors on reasonable request.

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

## Acknowledgements

We acknowledge Beat Blattmann and Céline Stutz-Ducommun of the Protein Crystallization Center UZH for performing the crystallization screening, and the staff of the SLS beamlines X06SA and X06DA for their support during X-ray data collection. STxB-Cy3 was obtained from Ludger Johannes (Curie Institute, Paris). Confocal laser scanning microscopy and imaging flow cytometry was performed using equipment of the Center of Microscopy and Image Analysis or the Flow Cytometry Facility, University of Zurich (UZH). SPR measurements were carried out at the Functional Genomics Center Zurich (FGCZ). Work in the group of H.H. was supported by the UZH, the Swiss National Science Foundation (SNF; 31003A_153200), the German Bundesministerium für Bildung und Forschung (BMBF; 031A410A; Infect-ERA project EUGENPATH), the Novartis Foundation for Medical-Biological Research, and the OPO foundation. M.A.S. was supported by a SNF Professorship (PP00P3_144823) and the UZH. A.W. was supported by a grant from the Swedish Research Council (2014-396). The funders had no role in the study design, data collection and analysis, decision to publish, or preparation of the manuscript.

## Author contributions

H.H. and M.A.S.: Conceived the study and designed experiments. K.B., C.A.J.H., A.L.S., B.S., A.W., M.H. and F.L.: Designed and performed experiments. K.B., M.A.S. and H.H.: Wrote the paper with contributions from all other authors.
