## [Peer Review file · Nature Communications]

Reviewers' comments:

Reviewer #2 (Remarks to the Author):

The authors established previously that RidL localizes to LCV membranes, likely through binding to phosphatidylinositol 4-phosphate and intervenes in retrograde trafficking by associating with the Vps29 subunit of the retromer. In this manuscript, they have identified the N-terminal domain of RidL, aa 2-281, as solely responsible for this interaction. They determined the crystal structure of this domain, which appears to have a new fold comprised mostly of alpha-helices, and identified two extensive loops at one end of the domain. Subsequently, they showed that deletion of one of these loops, named beta-hairpin loop, is essential for the interaction with Vps29 while deletion of the second loop has no effect on binding. They have shown further that the Ile170 is the hotspot of this interaction and a single mutation I170E abrogates the association with Vps29. Based on the recently determined structure of Vps29 with bound TBC1D5 GAP domain they investigated if RidL binds to the same hydrophobic region of Vps29 as the GAP domain and showed that, indeed L152E mutation in Vps29 in the center of this cavity is sufficient to block RidL binding. The experiments in vivo show that RidL can displace TBC1D5 on Vps29 preventing activation of Rab7 and the authors propose this displacement as the mechanism by which RidL affects the retrograde transport. The reported findings define the molecular mechanism of action of RidL and forward the understanding of Legionella infection mechanisms. The text is well written and the figures display the data supporting authors' conclusions.

Comments:

The authors describe the fold as 'unprecedented'. Based on DALI analysis this fold is not overly similar to those observed so far in other proteins (DALI always finds some similarities but they may have low Z-values and it would be good to know what was the highest Z and for what fragment) but my interpretation of the word 'unprecedented' is that a discovery of new fold is unheard of. This is of course not the case as new folds are still discovered albeit with low frequency. I suggest to use the word 'new fold' as better representing the structural reality.

The zooming on the L152E mutant of Vps29 on p. 8 should be better justified by mentioning here that this is the center of the surface to which the GAP domain binds and that is why it was investigated as a potential binding site for RidL.

p. 10, l. 9: "Intriguingly, the delta(beta)-hairpin ..." – this was already stated earlier so at this point it should not be a surprise but an awaited and expected behavior.

p. 11, l. 5-6: What restraint exactly was used in HADDOCK calculations? Are the conformations of binding loops of RidL and TBC1D5 similar in any way or completely different? Does RidL bind tighter than TBC1D5 as one would expect if it displays the latter?

p.12 l. 1: How was TBC1D5 identified in the cell?

p. 13, l. 2: " ... the mammalian homolog of TBC1D5 ..." – TBC1D5 refers to a human protein so this is not clear.

Fig. 2 legend: The data show RidL 9-258 domain, not the full-length protein. Since RidL binds phospholipids I would expect that it also localizes to membranes.

Fig. 3.: Showing elution profile would help to relate to the lanes in the SDS-PAGE. Why the band for Vps29 is so much weaker than RidL if they form 1:1 complex? Also, the complex would elute at earlier time than the RidL alone. The legend needs to be expanded.

Fig. 4 and the text p. 6: Is the hairpin loop solidly attached to the rest of the protein? Could one expect its movement away from the main body of RidL after binding to Vps29? This figure suggests that this could be a possibility. Any comments?

Reviewer #3 (Remarks to the Author):

In this manuscript, Bärlocher and colleagues have obtained structural information on the N-terminal portion of the Legionella pneumophila Type IV secretion effector RidL to gain insight into

its mode of action on the host retromer subunit Vps29. They convincingly show that a hydrophobic β -hairpin in RidL N-terminal portion binds a hydrophobic pocket in Vps29 that normally binds the Rab7 GAP TBC1D5. They further present evidence that RidL interaction with Vps29 displaces TBC1D5 from the retromer and from Legionella vacuoles during infection and that TBC1D5 contributes to bacterial replication.

Based on a structural biology approach, this study defines the molecular interactions and mode of action of a bacterial effector protein on its host target, advancing our understanding of how RidL operates to modulate retromer-associated functions. Interestingly, it proposes a mechanism by which displacement of a host factor may promote bacterial replication. These findings constitute an important conceptual advance in the field. The evidence presented to demonstrate the mechanism by which RidL binds Vps29 is compelling, yet the data arguing for TBC1D5 displacement and how this affects Legionella replication could benefit from stronger evidence.

Major comments:

1. An important aspect of this study deals with the consequences of RidL interaction with Vps29 on retromer function and Legionella replication. In particular, the authors present evidence based on information-driven docking analysis and fluorescence microscopy that RidL(9-258) displaces TBC1D5 from the retromer. Given the importance of these results to the study, it would be beneficial to further test for TBC1D5 displacement by RidL using purified proteins: since the retromer CSC-TBC1D5 interaction can be observed by GST pulldown (Jia et al 2016), showing that addition of RidL(9-258) in such an assay competes with retromer-TBC1D5 interaction would more convincingly demonstrate displacement.

2. Another concern is with the rather weak effect (20% increase) that RidL deletion or inactivation has on TBC1D5 recruitment to the Legionella vacuole. Could the analysis of TBC1D5 presence on vacuoles be plotted as individual GFP fluorescence values, instead of means, and would this reveal perhaps a stronger change in recruitment patterns than means can't show? How did the authors choose their time of analysis? Is that the peak of TBC1D5 recruitment? Additionally, the effect of TBC1D5 deletion in *D. discoideum* on bacterial replication appears quite limited. The fact that RidL may decrease TBC1D5 accumulation on the bacterial vacuole, yet this protein promotes replication is also counterintuitive. While the observed differences presented are statistically significant and appear consistent with the rest of the study, they look marginal and question the importance of the phenomenon, compared to the strong replication defect that a Δ RidL mutant displays.

3. With respect to marginal changes in replication, have the authors considered whether deletion of TBC1D5 could affect the kinetics of biogenesis of the Legionella replicative vacuole in a manner more dramatic than bacterial replication levels at 24 h?

Minor comments:

1. This is more out of curiosity, but I am wondering whether expressing RidL(9-258) in either amoebas or macrophages would affect vacuole biogenesis or replication of wild type and Δ RidL Legionella. Since it seems to exclude TBC1D5 from retromer complexes, yet does not inhibit retrograde traffic, it may provide useful information on the importance of TBC1D5 function vs inhibition of retrograde traffic in the context of infection.

2. Does ectopic expression of full length RidL displace TBC1D5 from the retromer?

3. page 8, lines 9-10: Please clarify the rationale for mutating Leu152 in Vps29, which I assume is because it has been shown to prevent Vps29-TBC1D5 interaction.

4. page 12, lines 4 and 9-10: It would be useful to add to Fig 5a GFP panels showing the

distribution of the various RidL constructs expressed.

5. page 13, line 19: The authors meant "...20% more TBC1D5..."

6. page 15, line 16: The statement claiming that "RidL removed TBC1D5 from LCVs" is quite an overinterpretation of the data, as the authors only see a 20% difference. This should be revised accordingly.

Response to reviewers

Reviewer #2:

The authors established previously that RidL localizes to LCV membranes, likely through binding to phosphatidylinositol 3-phosphate and intervenes in retrograde trafficking by associating with the Vps29 subunit of the retromer. In this manuscript, they have identified the N-terminal domain of RidL, aa 2-281, as solely responsible for this interaction. They determined the crystal structure of this domain, which appears to have a new fold comprised mostly of alpha-helices, and identified two extensive loops at one end of the domain. Subsequently, they showed that deletion of one of these loops, named beta-hairpin loop, is essential for the interaction with Vps29 while deletion of the second loop has no effect on binding. They have shown further that the Ile170 is the hotspot of this interaction and a single mutation I170E abrogates the association with Vps29. Based on the recently determined structure of Vps29 with bound TBC1D5 GAP domain they investigated if RidL binds to the same hydrophobic region of Vps29 as the GAP domain and showed that, indeed L152E mutation in Vps29 in the center of this cavity is sufficient to block RidL binding. The experiments in vivo show that RidL can displace TBC1D5 on Vps29 preventing activation of Rab7, and the authors propose this displacement as the mechanism by which RidL affects the retrograde transport.

The reported findings define the molecular mechanism of action of RidL and forward the understanding of *Legionella* infection mechanisms. The text is well written and the figures display the data supporting authors' conclusions.

Comments:

The authors describe the fold as 'unprecedented'. Based on DALI analysis this fold is not overly similar to those observed so far in other proteins (DALI always finds some similarities but they may have low Z-values and it would be good to know what was the highest Z and for what fragment) but my interpretation of the word 'unprecedented' is that a discovery of new fold is unheard of. This is of course not the case as new folds are still discovered albeit with low frequency. I suggest to use the word 'new fold' as better representing the structural reality.

Response: As suggested, we replaced "unprecedented fold" by "new fold" throughout the manuscript. The closest structural neighbour of RidL's N-terminal domain with a low Z-score of 4.8 and a high RMSD of 4.4 is PDB entry 1J1J, a so called human translin protein involved in RNA/DNA binding. This is now mentioned in the manuscript (p.5, l.18-20).

The zooming on the L152E mutant of Vps29 on p. 8 should be better justified by mentioning here that this is the center of the surface to which the GAP domain binds and that is why it was investigated as a potential binding site for RidL.

Response: We now justify focusing on the Vps29_{L152E} mutant by adding the statement: “Since the mutation Vps29_{L152E} specifically abolishes the interaction with host TBC1D5⁴¹, we further investigated this site for potential binding of RidL.” (p.9, l.1-2).

p. 10, l. 9: “Intriguingly, the delta(beta)-hairpin ...” – this was already stated earlier so at this point it should not be a surprise but an awaited and expected behaviour.

Response: We replaced “Intriguingly”, by “As expected” (p.10, l.16).

p. 11, l. 5-6: What restraint exactly was used in HADDOCK calculations? Are the conformations of binding loops of RidL and TBC1D5 similar in any way or completely different? Does RidL bind tighter than TBC1D5 as one would expect if it displaces the latter?

Response: As we specify in detail in the Materials and Methods section, we used the HADDOCK2.2 webserver (<http://milou.science.uu.nl/services/HADDOCK2.2/>) and ran the docking using the “Easy interface”, which only allows to define active and passive residue. As we further specify in the Materials and Methods, we only defined Ile₁₇₀ of RidL and Leu₂, Leu₂₅, Leu₁₅₂ and Tyr₁₆₅ of Vps29 as active residues (i.e. no passive residues were defined). As described (van Zundert GC, Bonvin AM. Modeling protein-protein complexes using the HADDOCK webserver "modeling protein complexes with HADDOCK". *Methods Mol Biol.* 2014;1137:163-79), HADDOCK2.2 carries out the following three steps: 1) rigid body docking, 2) semiflexible refinement and 3) flexible refinement in explicit solvent. The flexible refinements only resulted in minor changes (maximally 0.6 Å) of the Ca positions of the β-hairpin and in slight rearrangement (maximally 1.2 Å) of the side chains and preserved the side chain rotamers of the β-hairpin. The fact that docking only led to minor changes to the RidL structure is now mentioned in the manuscript (p.11, l.16-17).

Although the binding loops of RidL and TBC1D5 cover a similar region of Vps29, the interaction interface is not identical and the amino acid chains run in opposite directions. A shared hallmark is the interaction mediated by an isoleucine side chain, as highlighted in Fig. 4. With regard to binding loop conformations, in RidL the interaction is mediated by a highly constrained β-hairpin, while the TBC1D5 loop lacks the β-sheet interactions and is therefore presumably highly flexible unless bound to Vps29. Overall, the differences regarding the two binding loops appear quite speculative and are therefore not explicitly mentioned in the manuscript.

Our structural and docking analysis does not provide a solid basis to determine whether RidL can displace TBC1D5, because it is difficult to directly compare the results of the docking with the co-crystal structure containing the TBC1D5 peptide. In addition, a complex between full-length TBC1D5 and Vps29 may again reveal a somewhat different interaction mediated by this peptide sequence.

p.12 l. 1: How was TBC1D5 identified in the cell?

Response: Endogenous TBC1D5 was immuno-stained as described in Materials and Methods. This is now also mentioned in the text (p.12, l.10-11).

p. 13, l. 2: “... the mammalian homolog of TBC1D5 ...” – TBC1D5 refers to a human protein so this is not clear.

Response: The term “mammalian homolog of TBC1D5” was replaced by “mammalian TBC1D5” (p.13, l.20).

Fig. 2 legend: The data show RidL 9-258 domain, not the full-length protein. Since RidL binds phospholipids I would expect that it also localizes to membranes.

Response: The PtdIns(3)*P* binding site of RidL has not been localized to a specific region of RidL and thus, it was unknown whether (and how) the RidL 9-258 fragment localizes to membranes. To emphasize this fact, we added “binding of the fragment” to the title of the figure legend. The finding that translocated full-length RidL as well as the effector lacking the β -hairpin localizes to the poles of LCVs (Finsel et al., 2013; Fig. 6d), is in agreement with the presence of other membrane localization determinants in addition to the β -hairpin.

Fig. 3.: Showing elution profile would help to relate to the lanes in the SDS-PAGE. Why the band for Vps29 is so much weaker than RidL if they form 1:1 complex? Also, the complex would elute at earlier time than the RidL alone. The legend needs to be expanded.

Response: As suggested, we now show the elution profiles in Fig. 3 and Fig. S2. The intensity difference of the protein bands might be mainly due to the size differences of RidL (132kDa) and Vps29 (21kDa). Assuming an equal staining efficiency with Coomassie Brilliant Blue (CBB), a more than six-fold difference in favour of RidL would be expected at an equimolar ratio of the proteins. Moreover, we cannot exclude that RidL stains more efficiently with CBB than Vps29.

The elution peak of the RidL-Vps29 complex is actually shifted 0.2 ml earlier compared to RidL alone, which however, is almost beyond the resolution power of the SEC at a fraction size of 0.5 ml. An only small shift is expected due to the small size increase of the RidL-Vps29 complex compared to RidL alone.

Fig. 4 and the text p. 6: Is the hairpin loop solidly attached to the rest of the protein? Could one expect its movement away from the main body of RidL after binding to Vps29? This figure suggests that this could be a possibility. Any comments?

Response: As can be seen in Fig. 1b, the B-factors of the β -hairpin are comparatively low as compared with, e.g., the flexible F-loop. This suggests that the β -hairpin indeed might be solidly attached to remainder of the N-terminal domain. Moreover, the structure shows that there are multiple molecular interactions between the β -hairpin and the rest of RidL. These reflections are now included in the Discussion section (p.16, l.11-14).

Reviewer #3:

In this manuscript, Bärlocher and colleagues have obtained structural information on the N-terminal portion of the *Legionella pneumophila* type IV secretion effector RidL to gain insight into its mode of action on the host retromer subunit Vps29. They convincingly show that a hydrophobic β -hairpin in RidL N-terminal portion binds a hydrophobic pocket in Vps29 that normally binds the Rab7 GAP TBC1D5. They further present evidence that RidL interaction with Vps29 displaces TBC1D5 from the retromer and from *Legionella* vacuoles during infection and that TBC1D5 contributes to bacterial replication.

Based on a structural biology approach, this study defines the molecular interactions and mode of action of a bacterial effector protein on its host target, advancing our understanding of how RidL operates to modulate retromer-associated functions. Interestingly, it proposes a mechanism by which displacement of a host factor may promote bacterial replication. These

findings constitute an important conceptual advance in the field. The evidence presented to demonstrate the mechanism by which RidL binds Vps29 is compelling, yet the data arguing for TBC1D5 displacement and how this affects *Legionella* replication could benefit from stronger evidence.

Major comments:

1. An important aspect of this study deals with the consequences of RidL interaction with Vps29 on retromer function and *Legionella* replication. In particular, the authors present evidence based on information-driven docking analysis and fluorescence microscopy that RidL(9-258) displaces TBC1D5 from the retromer. Given the importance of these results to the study, it would be beneficial to further test for TBC1D5 displacement by RidL using purified proteins: since the retromer CSC-TBC1D5 interaction can be observed by GST pulldown (Jia et al 2016), showing that addition of RidL(9-258) in such an assay competes with retromer-TBC1D5 interaction would more convincingly demonstrate displacement.

Response: As suggested, we performed an additional *in vitro* experiment showing the interruption of GFP-TBC1D5-retromer interaction by purified RidL₁₀₋₂₅₈ in a β -hairpin dependent manner. HeLa cells transiently transfected with a plasmid encoding GFP-TBC1D5 were lysed, and GFP-TBC1D5 and associated retromer components were pulled down and washed. By adding purified RidL₁₀₋₂₅₈, RidL₁₀₋₂₅₈- $\Delta\beta$ -hairpin, or no protein, we demonstrated that RidL₁₀₋₂₅₈ displaces TBC1D5 from the retromer in a β -hairpin dependent manner. The experiment is documented in the new Fig. 5c and outlined in the text (p.12, l.22 - p.13, l.5).

2. Another concern is with the rather weak effect (20% increase) that RidL deletion or inactivation has on TBC1D5 recruitment to the *Legionella* vacuole. Could the analysis of TBC1D5 presence on vacuoles be plotted as individual GFP fluorescence values, instead of means, and would this reveal perhaps a stronger change in recruitment patterns than means can't show? How did the authors choose their time of analysis? Is that the peak of TBC1D5 recruitment? Additionally, the effect of TBC1D5 deletion in *D. discoideum* on bacterial replication appears quite limited. The fact that RidL may decrease TBC1D5 accumulation on the bacterial vacuole, yet this protein promotes replication is also counterintuitive. While the observed differences presented are statistically significant and appear consistent with the rest of the study, they look marginal and question the importance of the phenomenon, compared to the strong replication defect that a $\Delta ridL$ mutant displays.

Response: We would like to thank the reviewer for raising these valid points.

We agree that RidL has rather weak effects on TBC1D5 recruitment to LCVs. At this point, we cannot exclude that among the ~300 *L. pneumophila* effector proteins other toxins than RidL target the retromer or retromer-TBC1D5 interactions, which might compromise the effect(s) of RidL. These reflections are now outlined in the Discussion section (p.19, l.3-6).

As suggested, we plotted the individual data points of TBC1D5 fluorescence intensity on LCVs, but did not see a stronger effect. Therefore, we propose to keep the current way of data presentation (Fig. 6d; means and SD).

The time of analysis of TBC2D5 recruitment was chosen based upon infection assays using *D. discoideum* producing GFP-TBC1D5 in tandem with the endosomal marker AmtA-mCherry. We found that the amount of TBC1D5 on LCVs peaked at 2 h p.i., and accordingly, we used this time point for further analysis. This experiment is now shown in the new Fig. 6ab and outlined in the text (p.13, l.20 – p.14, l.4).

Furthermore, we agree that the finding that deletion of TBC1D5 actually reduces intracellular replication of *L. pneumophila* is rather counterintuitive. Previously, we made another seemingly paradox observation that the deletion of OCRL/Dd5P4 (a PtdIns(4,5) P_2 5-phosphatase producing PtdIns(4)P on LCVs, which *L. pneumophila* effectors bind to) indeed promotes intracellular growth of *L. pneumophila* (Weber *et al.*, 2009, Cell Microbiol 11: 442). TBC1D5 as well as OCRL likely adopt pleiotropic and complex functions in eukaryotic cells, which collectively define the role the host factor has during *L. pneumophila* infection. These reflections are now outlined in the Discussion section (p.19, l.13-19).

Finally, as outlined already previously in the Discussion (p.19, l.7-12), the 102 kDa C-terminal fragment of RidL likely harbors additional (perhaps catalytic) activities contributing to the inhibition of retrograde trafficking.

3. With respect to marginal changes in replication, have the authors considered whether deletion of TBC1D5 could affect the kinetics of biogenesis of the *Legionella* replicative vacuole in a manner more dramatic than bacterial replication levels at 24 h?

Response: To assess the kinetics and possible differences of LCV biogenesis in *D. discoideum* DH1 versus $\Delta tbc1d5$, we used the recruitment of calnexin to the pathogen vacuoles as a proxy. The quantitative analysis by imaging flow cytometry over an extended period of time during infection (15, 60, 120 min) did not reveal differences in pathogen vacuole formation. These data are now shown in the new Supplementary Fig. 4bc and outlined in the text (p.15, l.7-9).

Minor comments:

1. This is more out of curiosity, but I am wondering whether expressing RidL(9-258) in either amoebas or macrophages would affect vacuole biogenesis or replication of wild type and $\Delta ridL$ *Legionella*. Since it seems to exclude TBC1D5 from retromer complexes, yet does not inhibit retrograde traffic, it may provide useful information on the importance of TBC1D5 function vs inhibition of retrograde traffic in the context of infection.

Response: As suggested, we analyzed intracellular replication of *L. pneumophila* wild-type, $\Delta ridL$ or $\Delta icmT$ in *D. discoideum* ectopically producing RidL₁₋₂₅₈-GFP or RidL₁₋₂₅₈- $\Delta\beta$ -hairpin-GFP. We did not observe amoeba-dependent significant differences in intracellular growth for the different *L. pneumophila* strains. These data are shown in the new Supplementary Fig. 3 and outlined in the text (p.13, l.11-13).

2. Does ectopic expression of full length RidL displace TBC1D5 from the retromer?

Response: Due to poor expression of *ridL* in *D. discoideum* or HeLa cells, we recently constructed a eukaryotic codon-optimised variant. The production of RidL by the codon-optimized gene indeed led to a loss of the punctate localization of TBC1D5 and distribution to the cytosol. However, under these conditions, also Vps26 relocalized to the cytosol. Given this concomitant relocalization of Vps26 (and likely the other retromer subunits), the relocalization of TBC1D5 is no longer a proxy for TBC1D5-retromer dissociation. Further experiments using codon-optimized full-length RidL are ongoing, but we believe are beyond the scope of the current manuscript.

3. page 8, lines 9-10: Please clarify the rationale for mutating Leu152 in Vps29, which I assume is because it has been shown to prevent Vps29-TBC1D5 interaction.

Response (see also reviewer #2, comment 2): We now justify focusing on the Vps29_{L152E} mutant by adding the statement: “Since the mutation Vps29_{L152E} specifically abolishes the interaction with host TBC1D5⁴¹, we further investigated this site for potential binding of RidL.” (p.9, l.1-2).

4. page 12, lines 4 and 9-10: It would be useful to add to Fig 5a GFP panels showing the distribution of the various RidL constructs expressed.

Response: The GFP panels were added to Fig. 5a as suggested.

5. page 13, line 19: The authors meant “...20% more TBC1D5...”

Response: Thank you; we corrected this mistake (p.14, l.12).

6. page 15, line 16: The statement claiming that “RidL removed TBC1D5 from LCVs” is quite an overinterpretation of the data, as the authors only see a 20% difference. This should be revised accordingly.

Response: We replaced “removed TBC1D5 from LCVs” by “reduced TBC1D5 on LCVs” (p.17, l.7).

REVIEWERS' COMMENTS:

Reviewer #2 (Remarks to the Author):

The revised manuscript addressed satisfactorily all my comments.

Reviewer #3 (Remarks to the Author):

In this revised version of their manuscript, Barlocher and colleagues have thoroughly addressed and carefully responded to my original concerns with the study. In particular, my main concern about the displacement of TBC1D5 from the retromer complex by RidL has been satisfactorily addressed, and clarifications about the limited effect of RidL on TBC1D5 accumulation on LCV and TBC1D5's effect on bacterial replication have been made through additional experimental work. I do not have any further concerns about this well performed and significant study.

Minor comment:

Line 414: please remove "which might", which is duplicated in the sentence.